

**Long chain diols in settling particles in tropical oceans:**
**insights into sources, seasonality and proxies.**
Marijke W. de Bar*[,1], Jenny E. Ullgren[2], Robert C. Thunnell[‡], Stuart G. Wakeham[3], Geert-Jan
A. Brummer[4,5], Jan-Berend W. Stuut[4,5], Jaap S. Sinninghe Damsté[1,6] and Stefan Schouten[1,6]
[1] NIOZ Royal Netherlands Institute for Sea Research, Department of Marine Microbiology
and Biogeochemistry, and Utrecht University, P.O. Box 59, 1790 AB Den Burg, Texel, the
Netherlands
[2] Runde Miljøsenter, Runde, Norway
[3] Skidaway Institute of Oceanography, University of Georgia, 10 Ocean Science Circle,
Savannah, USA
[4] NIOZ Royal Netherlands Institute for Sea Research, Department of Ocean Systems, and
Utrecht University, P.O. Box 59, 1790 AB Den Burg, Texel, the Netherlands
[5] Vrije Universiteit Amsterdam, Faculty of Science, Department of Earth Sciences, De
Boelelaan 1085, 1081HV Amsterdam, the Netherlands
[6] Department of Earth Sciences, Faculty of Geosciences, Utrecht University, the Netherlands
* Corresponding author: Marijke W. de Bar (Marijke.de.Bar@nioz.nl)

[‡] Deceased: 30 July 2018.



**ABSTRACT**
In this study we have analyzed sediment trap time series from five tropical sites to assess seasonal
variations in concentrations and fluxes of long-chain diols (LCDs) and associated proxies with emphasis
on the Long chain Diol Index (LDI). For the tropical Atlantic, we observe that generally less than 2 %
of LCDs settling from the water column are preserved in the sediment. The Atlantic and Mozambique
Channel traps reveal minimal seasonal variations in the LDI, similar to the $TEX_{86}$ and $U^{K'}_{37}$. However,
annual mean LDI-derived temperatures are in good agreement with the annual mean satellite-derived
sea surface temperatures (SSTs). In the Cariaco Basin the LDI shows larger seasonal variation, as do
the $TEX_{86}$ and $U^{K'}_{37}$. Here, the LDI underestimates SST during the warmest months, which is likely due
to summer stratification and the habitat depth of the diol producers deepening to around 20 to 30 m.
Surface sediment LDI temperatures in the Atlantic and Mozambique Channel compare well with the
average LDI-derived temperatures from the overlying sediment traps, as well as with decadal annual
mean SST. Lastly, we observed large seasonal variations in the Diol Index, as indicator of upwelling
conditions, at three sites, potentially linked to Guinea Dome upwelling (Eastern Atlantic), seasonal
upwelling (Cariaco Basin) and seasonal upwelling and/or eddy migration (Mozambique Channel).



## 1. Introduction

Several proxies exist for the reconstruction of past sea surface temperature (SST) based on lipids. The $U^{K'}_{37}$ is one of the most applied proxies and is based on the unsaturation of long-chain alkenones (LCAs), which are produced by phototrophic haptophyte algae, mainly the cosmopolitan *Emiliania huxleyi* (Volkman et al., 1980; Brassell et al., 1986; Prahl and Wakeham, 1987; Conte et al., 1994). This index exhibits a strong positive correlation with SST (Müller et al., 1998; Conte, 2006). Another widely used organic paleotemperature proxy is the $TEX_{86}$, as originally proposed by Schouten et al. (2002), based on the relative distribution of archaeal membrane lipids, i.e. glycerol dialkyl glycerol tetraethers (GDGTs), and in the marine realm are mainly thought to be derived from the phylum Thaumarchaeota. Schouten et al. (2002) showed that the $TEX_{86}$ index measured in marine surface sediments is correlated with SST, and since then its application in paleoenvironmental studies has increased. However, research showed that despite the highest abundance of Thaumarchaeota in the upper 100 m of the water column, they can be present down to 5000 m depth (Karner et al., 2001; Herndl et al., 2005). Accordingly, GDGTs may be found in high concentrations below 100 m depth (e.g., Sinninghe Damsté et al., 2002; Wuchter et al., 2005) and several studies have indicated that $TEX_{86}$ might be more reflective of subsurface temperatures in some regions (e.g., Huguet et al., 2007; Lopes dos Santos et al., 2010; Kim et al., 2012; 2015; Schouten et al., 2013; Chen et al., 2014; Tierney et al., 2017; see Zhang and Liu, 2018 for review).

Most recently a SST proxy based on the distribution of long-chain diols (LCDs), called the Long-chain Diol Index, or LDI was proposed (Rampen et al., 2012). This index is a ratio of 1,13- and 1,15-diols (i.e., alcohol groups at position C-1 and C-13 or C-15), and the analysis of globally distributed surface sediments revealed that this index strongly correlates with SST. Since then, the index has been applied in several paleoenvironmental studies (e.g., Naafs et al., 2012; Lopes dos Santos et al., 2013; Jonas et al., 2017; Warnock et al., 2017). However, large gaps still remain in the understanding of this proxy. The largest uncertainty is that the main marine producer of LCDs is unknown. Although these diols have been observed in cultures of certain marine eustigmatophyte algae (e.g. Volkman et al., 1992; 1999; Méjanelle et al., 2003; Rampen et al., 2014b), the LCD distributions in cultures are different from those





observed in marine sediments. Furthermore, Balzano et al. (2018) combined lipid analyses with 18S
rRNA gene amplicon sequencing on suspended particulate matter (SPM) and did not find a significant
direct correlation between LCD concentrations and sequences of known LCD-producers. Rampen et al.
(2012) observed the strongest empirical relation between surface sediment derived LDI values and SSTs
for autumn to summer, suggesting that these are the main growth seasons of the source organisms.
Moreover, the strongest correlation was also observed for the upper 20 m of the water column,
suggesting that the LCDs are likely produced by phototrophic algae which thrive in the euphotic zone.
Nevertheless, LDI-temperatures based on surface sediments reflect an integrated signal of many years,
which complicates the interpretation of the LDI in terms of seasonal production and depth of export
production.
One way of resolving seasonality in LCD flux and LDI is to analyze time series samples from sediment
traps that continuously collect sinking particles in successive time intervals over periods of a year or
more. Such studies have been carried out for the $U^{K'}_{37}$ as well as for the $TEX_{86}$ and associated lipids
(e.g., Müller and Fischer, 2001; Wuchter et al., 2006; Huguet et al., 2007; Fallet et al., 2011; Yamamoto
et al., 2012; Rosell-Melé and Prahl, 2013; Türich et al., 2013). However, very few studies have been
done for LCDs. Villanueva et al. (2014) carried out a sediment trap study in Lake Challa (East Africa)
and Rampen et al. (2008) in the upwelling region off Somalia. The latter study showed that 1,14-diols,
produced by *Proboscia* diatoms strongly increased early in the upwelling season in contrast to 1,13- and
1,15-diols and thus can be used to trace upwelling. However, none of these sediment trap studies have
evaluated the LDI.
In this study, we assess seasonal patterns of the LDI for sediment trap series at five sites, i.e., in the
Cariaco Basin, the Mozambique Channel and three sites in the tropical North Atlantic and compared the
LDI values to satellite-derived SST, as well as results obtained for other temperature proxies, i.e. the
$TEX^{H}_{86}$ and $U^{K'}_{37}$. Moreover, for the Atlantic and Mozambique Channel, we compare the sediment trap
proxy signals with those preserved in the underlying sediments, after settling and burial. Finally, we
assess the applicability of the Diol Index, based on 1,14-diols produced by *Proboscia* diatoms
(Sinninghe Damsté et al., 2003), as tracer of upwelling and/or productivity in these regions.



**2. Materials and methods**

**2.1 Study sites and sample collection**

### 2.1.1    Tropical North Atlantic

The ocean current and wind patterns of the tropical Atlantic are mostly determined by the seasonal latitudinal shift of the intertropical convergence zone (ITCZ; Figure 1). The ITCZ migrates southward during boreal winter, and northward during boreal summer. During summer, the south-east trade winds prevail, whereas during winter the north-east trade winds intensify. The north-east trade winds drive the North Equatorial Current (NEC) which flows westward. South of this current flows the North Equatorial Countercurrent (NECC) towards the east (Stramma and Schott, 1999). The South Equatorial Current flows westward and branches off in the north Brazil Current (NBC; Stramma and Schott, 1999). When the ITCZ is in the north, the NBC retroflects off the South American coast, and is carried eastward into the NECC, and thus into the western tropical Atlantic (e.g., Richardson and Reverdin, 1987). North of the NBC, the Guiana Current (GC) disperses the outflow from the Amazon River towards the Caribbean Sea. (Müller-Karger et al., 1988; 1995). However, during boreal summer the NBC may retroflect, carrying the Amazon River plume far into the western Atlantic (e.g., Lefèvre et al., 1998; Müller-Karger et al., 1998; Coles et al., 2013). In fact, every late summer/autumn, the Amazon River outflow covers around $2 \times 10^6$ km$^2$ of the western North Atlantic, and the river delivers approximately half of all freshwater input into the tropical Atlantic (see Araujo et al., 2017 and references therein).

The eastern tropical North Atlantic is characterized by upwelling caused by the interaction between the trade winds and the movement of the ITCZ. Cropper et al. (2014) measured upwelling intensity along the NW African coastline between 1981 and 2012, in terms of wind speed, SST and other meteorological data. They recognized three latitudinal zones: weak permanent annual upwelling north of 26° N, strong permanent upwelling between 21° and 26° N and seasonal upwelling between 12° and 19° N related to the seasonal migration of the trade winds. Southeast of Cape Verde, large-scale cyclonic circulation forms the Guinea Dome (GD; Fig. 1), which centers around 10° N/22° W (Mazeika, 1967), i.e., close to mooring site M1. It is a thermal upwelling dome, formed by near-surface flow fields associated with the westward NEC, eastward NECC and the westward North Equatorial Undercurrent (NEUC) (Siedler et



al., 1992). It forms a cyclonic circulation as result of the eastward flowing NECC and the westward
flowing NEC (Rossignol and Meyrueis, 1964; Mazeika, 1967). The GD develops from late spring to
late fall due to the northward ITCZ position and the resulting Ekman upwelling, but shows significant
interannual variability (Siedler et al., 1992; Yamagata and Iizuka, 1995; Doi et al., 2009) judging from
general ocean circulation models. According to Siedler et al. (1992), upwelling is most intense between
July and October when the ITCZ is in the GD region and the NECC is strongest.
At three sites, we analyzed five sediment trap series along a latitudinal transect in the North Atlantic
(~12° N) to determine seasonal variations in the LDI. This transect has been studied previously for
Saharan dust deposition in terms of grain sizes (van der Does et al., 2016), as the tropical North Atlantic
receives approximately one third of the wind-blown Saharan dust (e.g., Duce et al., 1991; Stuut et al.,
2005), which might potentially act as fertilizer because of the high iron levels (e.g., Martin and
Fitzwater, 1988; Korte et al., 2017; Guirreiro et al., 2017; Goudie and Middleton, 2001 and references
therein). Furthermore, Korte et al. (2017) assessed mass fluxes and mineralogical composition,
Guerreiro et al. (2017) measured coccolith fluxes for two of the time series, while Schreuder et al.
(2018a; 2018b) measured long chain $n$-alkanes, long chain $n$-alkanols and fatty acids, and levoglucosan
for the same sediment trap samples and surface sediments as analyzed in this study.
At site M1 (12.00° N, 23.00° W), the sediment trap, referred to as M1U, was moored at a water depth
of 1150 m (Fig. 1). This mooring is located in the proximity of the Guinea Dome, and might therefore
potentially be influenced by seasonal upwelling. At station M2 (13.81° N, 37.82° W), two sediment
traps were recovered, i.e., an 'upper' (M2U) trap at a water depth of 1235 m, and a 'lower' (M2L) trap
at a depth of 3490 m. Lastly, at mooring station M4 (12.06° N, 49.19° W), also an upper and lower trap
series were recovered and analyzed (M4U and M4L), at 1130 and 3370 m depth, respectively. This
mooring site may seasonally be affected by Amazon River discharge (van der Does et al., 2016; Korte
et al., 2017; Guirreiro et al., 2017; Schreuder et al., 2018a). All sediment traps were equipped with 24
sampling cups, which sampled synchronously over 16-day intervals from October 2012 to November
2013, using HgCl$_2$ as a biocide and borax as a pH buffer to prevent in situ decomposition of the collected
material.

### 2.1.2   Mozambique Channel

The Mozambique Channel is located between Madagascar and Mozambique and is part of the Agulhas
Current system hugging the coast of South Africa (Lutjeharms, 2006). The Agulhas Current system is
an important conveyor in the transport of warm and salty waters from the Indian to the Atlantic Ocean
(Gordon, 1986; Weijer et al., 1999; Peeters et al., 2004). The northern part of the channel is also
influenced by the East African monsoon winds (Biastoch and Krauss, 1999; Sætre and da Silva, 1982;
Malauene et al., 2014). Between September and March, these winds blow from the northeast, parallel
to the Mozambique coastline, favoring coastal upwelling. Additionally, the Mozambique Channel is
largely influenced by fast-rotating, mesoscale eddies which migrate southward towards the Agulhas
region. Using satellite altimetry, Schouten et al. (2003) observed on average 4 to 6 eddies, ca. 300 km
in diameter, propagating yearly from the central Mozambique Channel (15° S) toward the Agulhas area
(35° S) between 1995 and 2000. Seasonal upwelling occurs off Northern Mozambique (between ca. 15
and 18° S) (Nehring et al., 1987; Malauene et al., 2014), from August to March with a dominant period
of about two months although periods of one to four weeks have also been observed (Malauene et al.,

161 2014).

The sediment trap was moored at 16.8° S and 40.8° E, at a water depth of 2250 m (Fig. 1; Fallet et al.,
2010, 2011) and of the same type as used for the North Atlantic transect. We analyzed the LCD proxies
for two respective time intervals: the first interval covers ca. 3.5 years, from November 2003 to
September 2007, with a sampling interval of 21 days. The second interval covers another year, between
February 2008 and February 2009, with a sampling interval of 17 days. Previously, Fallet et al. (2011)
published foraminiferal, $U^{K'}_{37}$ and $TEX_{86}$ records for the first time interval, and the organic carbon
content for the follow-up time series. For further details on the deployments and sample treatments, we
refer to Fallet et al. (2011, 2012). The two surface sediments are located across the narrowest transect
between Mozambique and Madagascar, and were analyzed for $U^{K'}_{37}$ and $TEX_{86}$ by Fallet et al. (2012)
and for LCDs by Lattaud et al. (2017b).



### 2.1.3   Cariaco Basin


The Cariaco Basin is one of the largest marine anoxic basins (Richards, 1975), located on the continental
shelf of Venezuela. The basin is characterized by permanent stratification and strongly influenced by
the migration of the intertropical convergence zone (ITCZ). During late autumn and winter, the ITCZ
migrates to the south which results in decreased precipitation and trade wind intensification which in
turn induces upwelling and surface water cooling. This seasonal upwelling is a major source of nutrients
that leads to strong phytoplankton growth along the Venezuelan coast (e.g., Müller-Karger et al., 2001;
Thunell et al., 2007). Between August and October, the ITCZ moves northward again, resulting in a
rainy season and diminishing of the trade winds inhibiting upwelling. During this wet season the
contribution of terrestrially derived nutrients is higher. Due to the prevalent anoxic conditions in the
basin, there is no bioturbation which has resulted in the accumulation of varved sediments which provide
excellent annually to decadally resolved climate records (e.g., Peterson et al., 1991; Hughen et al., 1996;
1998). Moreover, in November 1995, a time series experiment started to facilitate research on the link
between biogeochemistry and the downward flux of particulate material under anoxic and upwelling
conditions (Thunell et al., 2000). This project (CARIACO; http://imars.marine.usf.edu/cariaco)
involved hydrographic cruises (monthly), water column chemistry measurements and sediment trap
sampling (every 14 days). One mooring containing four automated sediment traps (Honjo and Doherty,
1988) was deployed at 10.50° N and 64.67° W, at a bottom depth of around 1400 m. These traps were
moored at 275 m depth, just above the oxic/anoxic interface (Trap A), 455 m (Trap B), 930 m (Trap C)
and 1255 m (Trap D). All traps contain a 13-cup carousel which collected sinking particles over 2 weeks,
and were serviced every half year. For further details on trap deployment and recovery, and sample
collection, storage and processing we refer to Thunell et al. (2000) and Goñi et al. (2004). In addition to
the sediment trap sampling, the primary productivity of the surface waters was measured every month
using [14]C incubations (Müller-Karger et al., 2001; 2004). For this study, we investigated two periods,
i.e., May 1999–May 2000 and July 2002–July 2003 for Traps A and B. These years include upwelling
and non-upwelling periods, as well as a disastrous flooding event in December 1999 (Turich et al.,
2013). Turich et al. (2013) identified the upwelling periods, linked to the migration of the ITCZ, as
indicated by decreasing SST in the CTD and satellite-based measurements (indicated by grey boxes in





figures 9 and 10), and shoaling of the average depths of primary production and increased primary
production. Moreover, Turich et al. (2013) evaluated the $U^{K'}_{37}$ and $TEX_{86}$ proxies for the same two time
series for which we analyzed the LCD proxies.

**2.2 Instrumental data**
Satellite SST, precipitation and wind speed time series of the M1, M2 and M4 moorings in the Atlantic
derive from Guerreiro et al. (2017 and in revision) who retrieved these data from the Ocean Biology
Processing Group (OBPG, 2014) (Frouin et al., 2003), the Goddard Earth Sciences Data and Information
Services Center (2016) (Huffman et al., 2007; Xie and Arkin, 1997) and NASA Aquarius project (2015a;
2015b) (Lee et al., 2012) (see supplement of Guerreiro et al., 2017 for detailed references). The SST and
Chlorophyll *a* time series data for the Mozambique Channel were adapted from Fallet et al. (2011), who
retrieved these data from the Giovanni database (for details see Fallet et al., 2011). Surface sediment
proxy temperatures were compared to annual mean SST estimates derived from the World Ocean Atlas
(2013) (decadal averages from 1955 to 2012; Locarnini et al., 2013). Sea surface temperature data for
the Cariaco Basin were adopted from Turich et al. (2013) and combined with additional CTD
temperatures from the CARIACO time series data base for the depths of 2, 5, 10, 15 and 20 m
(http://www.imars.usf.edu/CAR/index.html.; CARIACO time series composite CTD profiles; lead
principal investigator: Frank Müller-Karger).

**2.3 Lipid extraction**
**2.3.1 Tropical North Atlantic**
The 120 sediment trap samples were sieved through a 1 mm mesh wet-split into five aliquots (van der
Does et al., 2016), of which one was washed with Milli-Q water, freeze-dried and homogenized for
chemical analysis (Korte et al., 2017). For organic geochemistry, weight sub-aliquots were extracted as
described by Schreuder et al. (2018a). Shortly, ca. 100 mg dry weight of sediment trap residue, and
between 1.5 and 10 g of dry weight of surface sediment were extracted by ultrasonication using a mixture



of dichloromethane:methanol (DCM:MeOH) (2:1; v/v), and dried over a $Na_2SO_4$ column. For
quantification of LCDs, LCAs and GDGTs, we added the following internal standards to the total lipid
extracts (TLEs): 2.04 µg $C_{22}$ 7,16 diol (Rodrigo-Gamiz et al., 2015), 1.50 µg 10-nonadecanone ($C_{19:0}$
ketone) and 0.1 µg $C_{46}$ GDGT (Huguet et al., 2006), respectively. Subsequently, the TLEs were
separated into apolar (containing *n*-alkanes), ketone (containing LCAs) and polar (containing LCDs and
GDGTs) fractions over an activated (2h at 150 °C) $Al_2O_3$ column by eluting with hexane/DCM (9:1;
v/v), hexane/DCM (1:1; v/v) and DCM/MeOH (1:1; v/v), respectively. The apolar fractions were
analyzed by Schreuder et al. (2018a) for *n*-alkanes. Polar fractions were split for GDGT (25 %) and
LCD (75 %) analysis. The LCD fraction was silylated by the addition of BSTFA (*N,O*-
bis(trimethylsilyl)trifluoroacetamide) and pyridine, and heating at 60 °C for 20 min, after which ethyl
acetate was added prior to analysis. The ketone fraction was also dissolved in ethyl acetate, and the
GDGT fraction was dissolved in hexane:isopropanol (99:1, v/v) and analyzed by GC and GC/MS. Next,
the GDGT fractions were filtered through a 0.45 µm polytetrafluoroethylene (PTFE) filter and analyzed
by HPLC-MS.

**2.3.2 Mozambique Channel**

Aliquots of the sediment trap samples from the Mozambique Channel were previously extracted and
analyzed by Fallet et al. (2011) and Fallet et al. (2012), respectively. The sediment trap material was
extracted by ultrasonication using a mixture of DCM/MeOH (2:1; v/v), dried over $Na_2SO_4$, and
separated into apolar, ketone and polar fractions via alumina pipette column chromatography, by eluting
with hexane/DCM (9:1; v/v), hexane/DCM (1:1; v/v) and DCM/MeOH (1:1; v/v), respectively. These
existing polar fractions of the sediment trap material were silylated (as described above), dissolved in
ethyl acetate and re-analyzed for LCDs by GC-MS. Since no record was kept of the aliquoting of extracts
and polar fractions, we report the results in relative abundance rather than concentrations and fluxes of
diols.

**2.3.3 Cariaco Basin**

Sediment trap material was extracted as described by Turich et al. (2013). Briefly, 1/16 aliquots of the
trap samples were extracted by means of Bligh-Dyer extraction with sonication using a phosphate buffer



and a trichloroacetic acid (TCA) buffer, after which the extracts were separated by adding 5 % NaCl in
solvent-extracted distilled deionized water, and the organic phase was collected and the aqueous phase
was extracted two more times. The extracts were pooled and dried over Na$_2$SO$_4$ and separated by means
of Al$_2$O$_3$ column chromatography, eluting with hexane:DCM (9:1; v/v), DCM:MeOH (1:1; v/v) and
MeOH. For this study, this latter fraction was silylated (as described above), dissolved in ethyl acetate,
and analyzed for LCDs using GC-MS. Similar to the Mozambique Channel samples, no record was kept
of the aliquoting of extracts and polar fractions, and thus we report the results in relative abundance.

**2.4    Instrumental analysis**
**2.4.1 GDGTs**
The GDGT fractions of the surface sediments and sediment traps SPM samples of the tropical North
Atlantic were analyzed for GDGTs by means of Ultra High Performance Liquid Chromatography Mass
Spectrometry (UHPLC-MS). We used an Agilent 1260 HPLC, which is equipped with an automatic
injector, interfaced with a 6130 Agilent MSD, and HP Chemstation software according to Hopmans et
al. (2016). Compound separation was achieved by 2 silica BEH HILIC columns in tandem (150 mm x
2.1 mm; 1,7 µm; Waters Acquity) in normal phase, at 25 °C. GDGTs were eluted isocratically for 25
min with 18 % B, followed by a linear gradient to 35 % B in 25 minutes and finally a linear gradient to
100 % B in the last 30 min. A = hexane; B = hexane:isopropanol (9:1; v/v). The flow rate was constant
at 0.2 mL min$^{-1}$, and the injection volume was 10 µL. The APCI-MS conditions are described by
Hopmans et al. (2016). Detection and quantification of GDGTs was achieved in single ion monitoring
(SIM) mode of the protonated molecules ([M+H]$^+$) of the GDGTs. We used a mixture of crenarchaeol
and the C$_{46}$ GDGT (internal standard) to assess the relative response factor, which was used for
quantification of the GDGTs in the samples (c.f. Huguet et al., 2006).
Sea surface temperatures were calculated by means of the TEX$^H_{86}$ as defined by Kim et al. (2010), which
is a logarithmic function of the original TEX$_{86}$ index (Schouten et al., 2002):
$$\text{TEX}^H_{86} = \log \frac{[GDGT-2] + [GDGT-3] + [Cren']}{[GDGT-1] + [GDGT-2] + [GDGT-3] + [Cren']}$$    [1]



where the numbers indicate the number of cyclopentane moieties of the isoprenoid GDGTs, and *Cren´*
reflects an isomer of crenarchaeol, i.e. containing a cyclopentane moiety with a *cis* stereochemistry
(Sinninghe Damsté et al., 2018). The $\text{TEX}^H_{86}$ values were translated to SSTs using the core-top
calibration of Kim et al. (2010):
$$\text{SST} = 68.4 \times \text{TEX}^H_{86} + 38.6 \qquad [2]$$
The Branched Isoprenoid Tetraether (BIT) index is a proxy for the relative contribution of terrestrial
derived organic carbon (de Jonge et al., 2014; 2015). This ratio is based on the original index as proposed
by Hopmans et al. (2004), but includes the 6-methyl brGDGTs:
$$\text{BIT} = \frac{[brGDGT\ Ia] + [brGDGT\ IIa+IIa'] + [brGDGT\ IIIa+IIIa']}{[brGDGT\ Ia] + [brGDGT\ IIa+IIa'] + [brGDGT\ IIIa+IIIa'] + [Cren]} \qquad [3]$$
where the numbers reflect different branched GDGTs (see Hopmans et al., 2004) and *Cren* reflects
crenarchaeol. The branched GDGTs were always around the detection limit in the Atlantic samples,
implying a BIT index of around zero and thus minimal influence of soil organic carbon (Hopmans et al.,
2004), and thus the BIT index is not discussed any further.

**2.4.2  LCAs**
The ketone fractions of the surface sediments and sediment traps samples of the tropical North Atlantic
were analyzed for LCAs on an Agilent 6890N gas chromatograph (GC) with flame ionization
detection (FID) after dissolving in ethyl acetate. The GC was equipped with a fused silica column with
a length of 50 m, a diameter of 0.32 mm, and a coating of CP Sil-5 (film thickness = 0.12 μm). Helium
was used as carrier gas, and the flow mode was a constant pressure of 100 kPa. The ketone fractions
were injected on-column at a starting temperature of 70 °C, which increased by 20 °C min$^{-1}$ to 200 °C
followed by 3 °C min$^{-1}$ until the final temperature of 320 °C was reached. This end temperature was
held for 25 min.
The $\text{U}^{K'}_{37}$ index was calculated according to Prahl and Wakeham (1987):
$$\text{U}^{K'}_{37} = \frac{[C_{37:2}]}{[C_{37:2}] + [C_{37:3}]} \qquad [4]$$



The $U^{K'}_{37}$ values were translated to SST after the calibration of Müller et al. (1998):
$$SST = \frac{U^{K'}_{37} - 0.044}{0.033}$$   [5]
We have also applied the recently proposed BAYSPLINE Bayesian calibration of Tierney and Tingley
(2018). The authors showed that the $U^{K'}_{37}$ estimates substantially attenuate above temperatures of 24 °C,
moving the upper limit of the $U^{K'}_{37}$ calibration from approximately 28 to 29.6 °C at unity. Since our
traps are located in tropical regions with SSTs > 24 °C, we have applied this calibration as well.

### 2.4.3   LCDs

The silylated polar fractions were injected on-column on an Agilent 7890B gas chromatograph (GC)
coupled to an Agilent 5977A mass spectrometer (MS). The starting temperature was 70 °C, and
increased to 130 °C by 20 °C min$^{-1}$, followed by a linear gradient of 4 °C min$^{-1}$ to an end temperature of
320 °C, which was held for 25 min. 1μL was injected, and separation was achieved on a fused silica
column (25 × 0.32 mm) coated with CP Sil-5 (film thickness 0.12 μm). Helium was used as carrier gas
with a constant flow of 2 mL min$^{-1}$. The MS operated with an ionization energy of 70 eV. Identification
of LCDs was done in full scan mode, scanning between $m/z$ 50–850, based on characteristic
fragmentation patterns (Volkman et al., 1992; Versteegh et al., 1997). Proxy calculations and LCD
quantifications were performed by analysis in SIM mode of the characteristic fragments ($m/z$ 299, 313,
327 and 341; Rampen et al., 2012; $m/z$ 187 for internal diol standard). For quantification of LCDs in the
sediment traps and seafloor sediments of the tropical Atlantic, the peak areas of the LCDs were corrected
for the average relative contribution of the selected SIM fragments to the total ion counts, i.e., 16 % for
the saturated LCDs, 9 % for unsaturated LCDs and 25 % for the $C_{22}$ 7,16-diol internal standard.
Sea surface temperatures were calculated using the LDI index, according to Rampen et al. (2012):
$$LDI = \frac{[C_{30}\ 1,15-diol]}{[C_{28}\ 1,13-diol] + [C_{30}\ 1,13-diol] + [C_{30}\ 1,15-diol]}$$   [6]
These LDI values were converted into SSTs using the following equation (Rampen et al., 2012):





$$SST = \frac{LDI - 0.095}{0.033}$$    [7]
Upwelling conditions were reconstructed using the Diol Index as proposed by Rampen et al. (2008):
$$Diol\ Index = \frac{[C_{28}\ 1,14-diol] + [C_{30}\ 1,14-diol]}{[C_{28}\ 1,14-diol] + [C_{30}\ 1,14-diol] + [C_{30}\ 1,15-diol]}$$    [8]
In 2010, Willmott et al. introduced an alternative Diol Index, which is defined as the ratio of 1,14-diols
over 1,13-diols. Since the index of Rampen et al. (2008) includes the $C_{30}$ 1,15-diol, it can be affected by
temperature variation, and therefore we would normally prefer to use the index of Willmott et al. (2010).
However, we often did not detect the $C_{28}$ 1,13-diol, or it co-eluted with cholest-5-en-7-one-3β-ol,
compromising the calculation of the Diol Index of Willmott et al. (2010). Moreover, the temperature
variations in all three sediment traps are minimal as recorded by the LDI. Accordingly, we chose to
apply the Diol Index according to Rampen et al. (2008).
Potential fluvial input of organic carbon was determined by the fractional abundance of the $C_{32}$ 1,15-
diol (de Bar et al., 2016; Lattaud et al., 2017a):
$$F\mathrm{C}_{32}\ 1,15\text{-diol} = \frac{[C_{32}\ 1,15-diol]}{[C_{28}\ 1,13-diol] + [C_{30}\ 1,13-diol] + [C_{30}\ 1,15-diol] + [C_{32}\ 1,15-diol]}$$    [9]
The fractional abundance of the $C_{32}$ 1,15-diol was always lower than 0.23, suggesting low input of river
derived organic carbon (Lattaud et al., 2017a).

**2.5    Time-series analysis**
We performed time-series spectral analysis on the Diol Index data from the Mozambique Channel to
assess the influence of meso-scale eddies. Analyses were performed in MATLAB®. The two parts of
the Diol Index time series, i.e. the 2003–2007 and the 2008–2009 periods, were analysed both separately
and together. The data were linearly interpolated in time (to 21-day intervals for the 2003–2007 period,
and 17-day intervals for the 2008–2009 period) to adjust for disjunct sampling intervals or short gaps,
and detrended. A runs test for randomness (Gibbons & Chakraborty, 2003) showed that for the second,
shorter time series (2008–2009) the null hypothesis – that the values in the series are in random order –





could not be rejected at the 5 % significance level. The second series also lacked statistically significant
autocorrelation according to the Ljung-Box test (Ljung & Box, 1978). Therefore, there was little point
in analysing the shorter 2008–2009 time series for periodicity. We performed a wavelet analysis to detect
transient features in the Mozambique Channel Diol Index 2003–2007 time series following the methods
of Torrence and Compo (1998; http://paos.colorado.edu/research/wavelets/) and using the Morlet
wavelet as mother wavelet.

**3. Results**
**3.1 Tropical North Atlantic**
We have analyzed sediment trap samples from a latitudinal transect (~ 12°N) in the tropical North
Atlantic (two upper traps at ca. 1200 m water depth, and three lower traps at ca. 3500 m; Fig. 1), covering
November 2012–November 2013, as well as seven underlying surface sediments, for LCDs, LCAs and
GDGTs. Below we present the results for these lipid biomarkers and associated proxies.
**3.1.1 LCDs**
The LCDs detected in the sediment trap samples and surface sediments from the tropical North Atlantic
(Fig. 2) are the $C_{28}$ and (mono-unsaturated and saturated) $C_{30}$ 1,14- (between 1 and 49 % of all LCDs),
$C_{28}$ and $C_{30}$ 1,13- (0–3 %) and the $C_{30}$ 1,15- (44–99 %) and $C_{32}$ 1,15-diols (0–7 %). In the M2 and M4
traps, the $C_{30}$ 1,15-diol constitutes between 87 and 95 % of total LCDs. We detected the $C_{29}$-OH fatty
acid in the traps from M1 and M4, in a few samples of the M2 traps and in all surface sediments.
Similarly, the $C_{28}$ 1,14-diol was detected in all samples from M1 and M4, in only a few M2 samples and
in all surface sediments. For most samples from M2U and M2L, the $C_{28}$ 1,14-diol was often part of a
high background signal, making identification and quantification problematic. In these cases, 1,14-diol
fluxes and Diol Index were solely based on the (saturated and mono-unsaturated) $C_{30}$ 1,14-diol. In
contrast, the saturated $C_{30}$ 1,14-diol was detected in all samples.
The average [1,13+1,15]-diol flux is 2.6 ($\pm$ 1.0) µg m$^{-2}$ d$^{-1}$ at M1U, 1.4 ($\pm$ 1.2) and 1. 2 ($\pm$ 1.1) µg m$^{-2}$ d$^{-}$
$^1$ for M2U and M2L, respectively, and 7.0 ($\pm$ 7.8) and 2.2 ($\pm$ 3.3) µg m$^{-2}$ d$^{-1}$ for M4U and M4L,



respectively (Fig. 3). The [1,13+1,15]-diol and 1,14-diol concentrations in the underlying sediments
vary between 0.05 µg g$^{-1}$ and 0.50 µg g$^{-1}$, and between 3 ng g$^{-1}$ and 0.06 µg g$^{-1}$, respectively. The
[1,13+1,15]-LCD flux is more than three times higher in the upper trap of M4 than in the lower trap,
whereas at M2, where the average LCD fluxes are much lower, the difference is not appreciable. The
1,14-diol flux for M1U averages 0.5 (± 0.8) µg m$^{-2}$ d$^{-1}$ with a pronounced maximum of 3.5 µg m$^{-2}$ d$^{-1}$ in
late April (Fig. 6a), irrespective of the total mass flux. The average 1,14-diol flux at M2 is much lower
and similar for the upper and lower traps, being around 0.01–0.02 (± 0.01) µg m$^{-2}$ d$^{-1}$. At M4, the average
1,14-diol fluxes are 0.3 (± 0.5) and 0.1 (± 0.2) µg m$^{-2}$ d$^{-1}$ for the upper and lower trap, respectively.
There are two evident maxima in the [1,13+1,15]-diols and 1,14-diol fluxes in late April and during
October/November, concomitant with maxima in the total mass flux (Fig. 3d and 3e). However, in the
lower trap this flux maximum is distributed over two successive trap cups, corresponding to late
April/early May (Fig. 3e and 3j).
The LDI ranged between 0.95 and 0.99 in all traps, corresponding to temperatures of 26.0 to 27.3 °C
with no particular trends (Fig. 5). For most M2 and M4 samples the C$_{28}$ 1,13-diol was below
quantification limit and, hence, LDI was always around unity, corresponding to 26.9 to 27.3 °C (Fig. 5),
whereas in others samples the C$_{28}$ 1,13-diol co-eluted with cholest-5-en-7-one-3β-ol, prohibiting the
calculation of the LDI and Diol Index (Fig. 5 and 6). The flux-weighted annual average LDI-derived
SSTs are 26.6 °C for M1U, and 27.1 °C for M2U, M2L, M4U and M4L. The underlying sediment is
very similar, with LDI values between of 0.95 and 0.98 corresponding to 26.0 and 26.9 °C. The Diol
Index varied from 0.03 to 0.30 in M1U, showing a pronounced maximum during spring (Fig. 6a). The
Diol Index at M2 ranges between 0.01 and 0.05 without an evident pattern, while the Diol Index at M4
ranges from 0.01 to 0.10 and shows the same pattern in the lower and upper trap, with highest values
during spring (ca. 0.1), followed by a gradual decrease during summer (Fig. 6d; 6e).

### 3.1.2 LCAs

We detected C$_{37}$, C$_{38}$ and C$_{39}$ long-chain alkenones in the sediment trap and surface sediments. The C$_{37:3}$
alkenone was generally around the limit of quantification for the M2L and M4L traps, and below the





limit of quantification for 4 out of the 7 surface sediment samples, while the $C_{37:2}$ alkenone was always
sufficiently abundant. The annual mean fluxes of the $C_{37}$ LCAs are 4.3 ($\pm$ 3.5) µg m$^{-2}$ d$^{-1}$ for M1U, 1.2
($\pm$ 0.9) µg m$^{-2}$ d$^{-1}$ and 0.4 ($\pm$ 0.2) µg m$^{-2}$ d$^{-1}$ for M2U and M2L, respectively, and 2.8 ($\pm$ 5.0) µg m$^{-2}$ d$^{-1}$
and 1.2 ($\pm$ 2.0) µg m$^{-2}$ d$^{-1}$ for M4U and M4L, respectively. The concentrations of the $C_{37}$ LCAs in the
underlying surface sediments range between 0.02 and 0.41 µg g$^{-1}$. At M4, the two total mass flux peaks
at the end of April and during October/November are also clearly pronounced in the $C_{37}$ alkenone fluxes
(Fig. 3d, 3e and 6g), as well as the increased signal in the cup reflecting the beginning of May, which
follows the cup which recorded the peak in total mass flux at the end of April. The $U^{K'}_{37}$ varied from
0.87 to 0.93, corresponding to 25.1 to 27.0 °C (Fig. 7c) for 3 out of 7 surface sediments in which the
$C_{37:3}$ was above quantification limit. The flux-weighted average SSTs are 26.1 °C for M1U, 25.7 and
26.4 °C for M2U and M2L, respectively, and 28.2 and 27.5 °C for M4U and M4L, respectively (Fig. 7).
SST variations per sediment trap are generally within a 2–3 °C range (Fig. 5) with no apparent trends.

**3.1.3 GDGTs**

The main GDGTs detected were the isoprenoidal GDGT-0, -1, -2, -3, crenarchaeol and the isomer of
crenarchaeol. Branched GDGTs were typically around or below quantification limit. Additionally, we
detected three hydroxyl GDGTs (OH-GDGTs), i.e. OH-GDGT-0, -1 and -2. These OH-GDGTs
contributed ca. 0.1–0.2 % to the total GDGT pool (i.e., hydroxyl and isoprenoidal) in the sediment traps,
but in the surface sediments their fractional abundance was higher, around 1 %. The average iGDGT
flux in M1U is 15.5 ($\pm$ 4.6) µg m$^{-2}$ d$^{-1}$, 2.4 ($\pm$ 1.1) and 2.6 ($\pm$ 0.3) µg m$^{-2}$ d$^{-1}$ in M2U and M2L,
respectively, and 4.3 ($\pm$ 1.5) and 2.9 ($\pm$ 1.2) µg m$^{-2}$ d$^{-1}$ in M4U and M4L, respectively (Fig. 3f). The
surface sediments exhibit iGDGT concentrations between 0.4 and 1.7 µg g$^{-1}$. Sediment $TEX^{H}_{86}$ values
vary between 0.62 and 0.69, corresponding to 24.3 to 27.4 °C. The $TEX^{H}_{86}$ flux-weighted average SSTs
are 25.2 °C for M1U, 27.3 and 26.6 °C for M2U and M2L, respectively, and 27.8 and 26.7 °C for M4U
and M4L, respectively. SSTs vary typically within a range of 1 and 2 °C. At M2U and M4U, the $TEX^{H}_{86}$
temperatures decrease slightly (ca. 1–2 °C) during January and July (Fig. 5b and 5d).

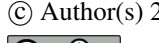



### 3.2 Mozambique Channel

For two time series (November 2003–September 2007 and February 2008–February 2009), we have analyzed LCDs collected in the sediment trap at 2250 m water depth as well as nearby underlying surface sediments (Fig. 1). The main LCDs observed in the sediment traps and surface sediments are the $C_{28}$ 1,12-, 1,13- and 1,14-diols, the $C_{30}$ 1,13-, 1,14- and 1,15-diols and the $C_{32}$ 1,15-diol. We also observed the $C_{30:1}$ 1 1,14 diol in some trap samples, and the $C_{29}$ 12-OH fatty acid in all trap and sediment samples. The $C_{30}$ 1,15 is generally highest in abundance, varying between 28 and 85 % of the total LCD assemblage. The $C_{28}$ and $C_{30}$ 1,14-diols contribute between 11 and 67 % of total LCDs. In 24 samples, the $C_{28}$ 1,13-diol co-eluted with cholest-5-en-7-one-3β-ol, and henceforth we did not calculate the LDI for these samples. The LDI varied between 0.94 and 0.99, i.e., close to unity, corresponding to 25.5 to 27.2 °C, without an evident trend (Fig. 8a). The Diol Index ranges between 0.11 and 0.69, showing substantial variation, although not with an evident trend (Fig. 8b). The average LDI-derived temperature of two underlying surface sediments is 26.0 °C.

### 3.3 Cariaco Basin

We analyzed LCDs for two time series (May 1999–May 2000 and July 2002–July 2003) from the upper (Trap A; 275 m) and the lower trap (Trap B; 455 m) in the Cariaco Basin. The main LCDs detected for both time series are the $C_{28}$ 1,14-, $C_{30}$ 1,14-, $C_{30:1}$ 1,14-, $C_{28}$ 1,13-, $C_{30}$ 1,15- and $C_{32}$ 1,15-diols, as well as the $C_{29}$ 12-OH fatty acid. The $C_{30}$ 1,15-diol contribution varies between 3 and 92 % of all LCDs, the $C_{28}$ and $C_{30}$ 1,14-diol contribution between 3 and 96 %, and the $C_{28}$ and $C_{30}$ 1,13-diols constitute between 0 and 8 %. For some samples we did not compute the LDI, as the $C_{28}$ 1,13-diol co-eluted with cholest-5-en-7-one-3β-ol. The calculated LDI values range between 24.3 and 25.3 °C and 22.0 and 27.2 °C for Trap A and B of the 1999-2000 time series, respectively, with the lowest temperature during winter, and the highest during summer. For the 2002-2003 time series, LDI temperatures for Trap A range between 23.3 and 26.2 °C, and for Trap B between 22.5 °C and 26.5 °C.



For the May 1999–May 2000 time series, the Diol Index varies between 0.05 and 0.97 for Trap A, and
between 0.05 and 0.91 for Trap B (Fig. 9) with similar trends, i.e. the lowest values of around 0.1-0.2
just before the upwelling period during November, rapidly increasing towards values between ca. 0.8
and 1 during the upwelling season (January and February). For the time series of July 2002–July 2003,
the Diol Index shows similar trends, i.e. Diol Index values around 0.8-0.9 during July, which rapidly
decrease towards summer values of around 0.2-0.3. Similar to the 1999-2000 time series, the lowest
index values (ca. 0.2) are observed just before the upwelling period (during September), after which
they increase towards values of around 0.8-0.9 between December and March at the start of the
upwelling season. At the end of the upwelling season the Diol Index increases, followed by another
maximum of around 0.6 during May.
**4. Discussion**
**4.1 LCD sources and seasonality**
The 1,14 diols can potentially be derived from two sources, i.e. *Proboscia* diatoms (Sinninghe Damsté
et al., 2003; Rampen et al., 2007) or the dictyochophyte *Apedinella radians* (Rampen et al., 2011). The
non-detection of the $C_{32}$ 1,14-diol, which is a biomarker for *Apedinella radians* (Rampen et al., 2011),
and the detection of the $C_{30:1}$ 1,14 diol and $C_{29}$ 12-OH fatty acid, which are characteristic of *Proboscia*
diatoms (Sinninghe Damsté et al., 2003), suggests that *Proboscia* diatoms are most likely the source of
1,14-diols in the tropical North Atlantic, the Mozambique Channel and the Cariaco Basin.
In the Cariaco Basin, the Diol Index shows a strong correlation with primary production rates,
suggesting that *Proboscia* productivity was synchronous with total productivity (Fig. 9). Primary
productivity in the Cariaco Basin is largely related to seasonal upwelling which occurs between
November and May when the ITCZ is at its southern position. Hence, the Diol Index seems to be an
excellent indicator of upwelling intensity in the Cariaco Basin.
The index also shows considerable variation over time in the Mozambique Channel (Fig. 8b). Previous
studies have shown that upwelling occurs in the Mozambique Channel between ca. 15 and 18° (Nehring
et al., 1987;  Malauene et al., 2014), i.e. at the location of our sediment trap. Upwelling is reflected by



cool water events and slightly enhanced Chlorophyll *a* levels, and Malauene et al. (2014) observed cool
water events at ca. two month intervals although periods of 8 to 30 days were also observed. The two
main potential forcing mechanisms for upwelling in the Mozambique Channel are the East African
monsoon winds and the meso-scale eddies migrating through the channel. Fallet et al. (2011) showed
that subsurface temperature, current velocity and the depth of surface-mixed layer all revealed a
dominant periodicity of four to six cycles per year, which is the same frequency as that of the southward
migration of meso-scale eddies in the channel (Harlander et al., 2009; Ridderinkhof et al., 2010),
implying that eddy passage strongly influences the water mass properties. Wavelet analysis of the Diol
Index for the period 2003–2007 (not shown) revealed short periods, occurring around January of 2004,
2005, and 2006, of significant (above the 95 % confidence level) variability at about bimonthly
frequencies (60-day period). Both the frequency and the timing of the observed time periods of enhanced
Diol Index variability are similar to those of the cool water events as observed by Malauene et al. (2014),
associated with upwelling (Fig. 8b). The strongest variability of the Diol Index at frequencies of four
cycles per year and higher occurred in the first half of 2006. During the same period, salinity time series
showed the passage of several eddies that had a particularly strong effect on the upper layer hydrography
(Ullgren et al., 2012). Malauene et al. (2014) showed that neither upwelling-favorable winds, nor
passing eddies, can by themselves explain the observed upwelling along the northern Mozambique
coast. The two processes may act together, and both strongly influence the upper water layer and the
organisms living there, potentially including the LCD producers.
The least (seasonal) variation in the Diol Index is observed at M2 in the tropical North Atlantic (Fig. 6b
and 5c), which is likely due to its central open ocean position, associated with relatively stable,
oligotrophic conditions (Guerreiro et al., 2017). In contrast, M4 and M1 are closer to the south American
and west African coast, respectively, and thus are potentially under the influence of Amazon river runoff
and upwelling, respectively, and specific wind and ocean circulation regimes (see Sect. 2.1.1). However,
at M4, the Diol Index is also low (max. 0.1), suggesting low *Proboscia* productivity (Fig. 6d and 5e).
At M1, by contrast, we observe enhanced values for the Diol Index of up to ~0.3 during spring (Fig. 6a).
Most likely, an upwelling signal at this location is associated with the seasonal upwelling of the Guinea





Dome. This upwelling is generally most intense between July and October (Siedler et al., 1992), due to the northward movement of the ITCZ and the resulting intensified Ekman upwelling. Specifically, during this period, the trade winds are weaker, atmospheric pressure is lower, and the regional wind stress is favorable to upwelling of the North Equatorial Undercurrent (Voituriez, 1981). Indeed, a decrease in wind speed and increased precipitation during summer to autumn was observed (Fig. 6a) which confirms that during these seasons the ITCZ was indeed at a northern position, and that during 2013 the upwelling associated with the Guinea Dome was most favored between July and October. The timing of the Diol Index peak, i.e., between March and June is consistent with previous sediment trap studies elsewhere which have shown that *Proboscia* diatoms and 1,14-diols are typically found during pre-upwelling or early upwelling periods (Koning et al., 2001; Smith, 2001; Sinninghe Damsté et al., 2003; Rampen et al., 2007). The surface sediment at 22° W just east of M1 also reveals the highest Diol Index (0.53), likely due its closer vicinity to the Guinea Dome center. Several studies have reported *P. alata* diatoms offshore NW Africa (Lange et al., 1998; Treppke et al., 1995; Crosta et al., 2012; Romero et al., 1999), pointing to *P. alata* as a plausible source organism. The sedimentary annual diol indices compare well with the sediment trap indices (Fig. 7e), which is consistent with the results of Rampen et al. (2008).

To assess variations in seasonal production of 1,13- and 1,15-diols in the tropical Atlantic, for which we have the most complete dataset, we calculated the flux-weighted 1,13- and 1,15-diol concentrations for the different traps, and summed these per season (Fig. 4). Highest production is observed in autumn, followed by summer and spring, with the lowest production during winter (~60 % compared to autumn). This is in agreement with Rampen et al. (2012) who observed, for an extensive set of surface sediments, the strongest correlation between LDI and SST for autumn, suggesting that production of the source organisms of the LDI mainly occurs during autumn. At M4, there are two evident peaks in the 1,13- and 1,15-diol fluxes at the end of April and October 2013. These maxima correlate with peaks in other lipid biomarker fluxes (i.e., 1,14-diols, $C_{37}$ alkenones and iGDGTs), total mass flux, calcium carbonate ($CaCO_3$), OM and the residual mass flux which includes the deposition flux of Saharan dust (Korte et al., 2017). According to Guerreiro et al. (2017), the maximum in total mass flux at the end of April 2013





is likely caused by enhanced export production due to nutrient enrichment as a result of wind-forced
vertical mixing. The peak at the end of October 2013, is likely associated with discharge from the
Amazon River. Moreover, both peaks are concomitant with prominent dust flux maxima, suggesting
that Saharan dust also acted as nutrient fertilizer (Korte et al., 2017; Guerreiro et al., 2017). Guirreirro
et al. (2017) suggested that during the October-November event the Amazon River may not only have
acted as nutrient supplier, but also as buoyant surface density retainer of dust-derived nutrients in the
surface waters, resulting in the development of algal blooms within just a few days, potentially
explaining the peak 1,13- and 1,15-diol fluxes, as well as the peak fluxes of the other lipid biomarkers.
However, they might also partially result from enhanced particle settling, caused by e.g. dust ballasting
or faecal pellets of zooplankton (see Guerreiro et al. 2017 and references therein). This agrees with the
results of Schreuder et al. (2018a) who show that the $n$-alkane flux also peaks concomitant with the
peaks in total mass flux and biomarkers, whereas $n$-alkanes are terrestrial derived (predominantly
transported by dust) and increased deposition can therefore not result from increased primary
productivity in the surface waters.
The $C_{37}$ alkenone flux at M4U also reveals these two distinct maxima at the end of April and October
during 2013 (Fig, 6g). Interestingly, this flux, as well as the alkenone flux at M2U, is consistent with
coccolith export fluxes of the species *Emiliania huxleyi* and *Gephyrocapsa oceanica* (Guerreiro et al.,
2017). In fact, when we combine the coccolith fluxes of both species, we observe strong correlations
with the $C_{37}$ alkenone fluxes for both M2U and M4U (Fig. 6f and 6g, respectively; $R^2 = 0.60$ and 0.84
for M2U and M4U, respectively). This implies that these two species are the main LCA producers in
the tropical North Atlantic, which agrees with previous findings (e.g., Marlowe et al., 1984; Brassell,
2014; Conte et al., 1994; Volkman et al., 1995).

**4.2 Preservation of LCDs**

The sediment trap data from the North Atlantic can be used to assess the relative preservation of LCDs,
as well as other proxy lipid biomarkers, by comparing the flux-weighted concentration in the traps with
the concentrations in the surface sediments. For all four biomarker groups, i.e., $C_{37}$ alkenones, iGDGTs,



1,14-diols and 1,13- and 1,15-diols, we observe that in general the flux-weighted concentrations are
higher in the upper traps (ca. 1200 m) as compared to the lower traps (ca. 3500 m; Fig. 2) by a factor of
between 1.2 and 4.4, implying degradation during settling down the water column. The concentrations
in the surface sediments are 2 to 3 orders of magnitude lower in concentration (i.e., between 0.1–1.5 %
of upper trap signal), implying that degradation of lipids is mainly taking place at the water-sediment
surface rather than the water column. A similar observation was made for levoglucosan in these sediment
traps (Schreuder et al., 2018b). This is likely linked to the extent of the oxygen exposure time (Hartnett
et al., 1998; Hedges et al., 1999) at the seafloor (Hartnett et al., 1998; Sinninghe Damsté et al., 2002),
since during settling the lipids are exposed to oxygen for weeks, whereas for surface sediments this is
typically decades to centuries. Our results compare well with several other sediment trap studies which
showed that LCDs, LCAs and iGDGTs generally have a preservation factor of around 1 % (surface
sediment vs. trap) (e.g., Prahl et al., 2000; Wakeham et al., 2002; Rampen et al., 2007; Yamamoto et al.,

2012).

We have also identified the $C_{30}$ and $C_{32}$ 1,15-keto-ol for in the Atlantic as well as the Mozambique and
Cariaco sediment traps and surface sediments. These lipids are structurally related to LCDs and occur
ubiquitously in marine sediments (e.g., Versteegh et al., 1997; 2000; Bogus et al., 2012; Rampen et al.,
2007; Sinninghe Damsté et al., 2003; Wakeham et al., 2002; Jiang et al., 1994), and were inferred to be
oxidation products of LCDs (Ferreira et al., 2001; Bogus et al., 2012; Sinninghe Damsté et al., 2003).
We have not detected 1,14-keto-ols, which supports the hypothesis of Ferreira et al. (2001) and
Sinninghe Damsté et al. (2003) that the silica frustules of *Proboscia* diatoms sink relatively fast and thus
are exposed to oxygen for a shorter period than the 1,13- and 1,15-diols, and thus less affected by
oxidation.
For both the tropical Atlantic and the Cariaco Basin, we observe highly similar LDI values for the upper
and the lower traps. In the Atlantic there is no statistical difference between upper and lower trap that
are 2200 m apart (two-tailed $p > 0.8$), but we have too little data for the Cariaco Basin for statistical
comparison (Fig. 7b, 9c and 9f). This suggests that degradation in the water column does not affect the
LDI proxy. This is in agreement with the study of Reiche et al. (2018) who performed a short-term



degradation experiment (< 1 year) and found that the LDI index was not affected by oxic exposure on
short time scales. However, the oxygen exposure time on the seafloor is much longer, and Rodrigo-
Gámiz et al. (2016) showed for sediments in the Arabian Sea, deposited under a range of bottom water
oxygen conditions, that different LCDs had different degradation rates, which compromised the LDI
ratio. For the three sites in the tropical North Atlantic, we have calculated the flux-weighted average
proxy values for every sediment trap and compare these with the underlying surface sediments (Fig. 7b-
7e). For all indices, i.e., Diol Index, LDI, $U^{K'}_{37}$ and $TEX_{86}$, we observe very good correspondence
between the sediment trap and surface sediment values, implying minimal alteration of the proxies after
settling and during burial. Similarly, for the Mozambique Channel, the mean Diol Index and LDI from
the sediment trap (i.e., 0.41 and 0.97, respectively) are very similar to the surface sediment values (i.e.,
0.42 and 0.95, respectively). In agreement with the consistent diol indices, we observe that all individual
LCDs are also preserved relatively equally in the tropical Atlantic (1.2-4.3 % at station M1, 0.1-2.9 %
at station M2 and 0.03-0.16 % at station M4). This contrasts with the findings of Rodrigo-Gámiz et al.
(2016) who found that the 1,15-diols have the highest degradation rate, followed by the 1,14- and 1,13-
diols. Only the $C_{32}$ 1,15-diol seems relatively better preserved than the other LCDs at all three North
Atlantic mooring sites (Fig. 2), suggesting that the $C_{32}$ 1,15-diol is less impacted by degradation. The
$C_{32}$ 1,15-diol likely partially derives from the same source as the other 1,13- and 1,15-diols, but is also
produced in fresh water systems (e.g., Versteegh et al., 1997; 2000; Rampen et al., 2014b; de Bar et al.,
2016; Lattaud et al., 2017a; 2017b). Hence, the different preservation characteristics might be the result
of a different source for this LCD.

**4.3 Relationship between LDI and SST**

In the tropical Atlantic and Mozambique Channel, the LDI-derived SSTs show minimal differences (<2
°C), while in the Cariaco Basin we observe much larger changes that range from 22.0 °C to 27.2 °C
(Fig. 9). Both time series in the Cariaco Basin show low temperatures between November and May
associated with the seasonal upwelling and surface water cooling, and significantly higher temperatures
during the rainy summer. However, during the warmest periods, the LDI temperatures are generally





lower than measured at the surface by CTD, whereas during the colder phases, the LDI agrees well with
the measurements. The LDI calibration reaches unity at 27.4 °C, and therefore it is not possible to resolve
the highest temperatures which are between ca. 28 and 30 °C. However, the LDI-derived temperatures
are sometimes well below 27.4 °C where the CTD data suggest SSTs > 28 °C. Consequently, the LDI-
based temperatures agree with CTD-based SSTs within calibration error for most of the record, but
during summer when SST is highest, are offset outside the calibration error ($\Delta T$ ~2.5-4.5 °C).
Interestingly, the $U^{K'}_{37}$- and $TEX^H_{86}$-derived temperature trends show the same phenomenon (Turich et
al., 2013; Fig. 9), where the proxy temperatures are cooler than the measured temperatures during the
warmer months. For $U^{K'}_{37}$, Turich et al. (2013) pointed out that a time lag between synthesis, export and
deposition could potentially explain the difference between the proxy and CTD temperatures. However,
previous analysis of plankton biomass, primary productivity, bio-optical properties and particulate
organic carbon fluxes for the same time period (Müller-Karger et al., 2004), as well as the total mass
and terrigenous fluxes assessed by Turich et al. (2013) showed best correlation at zero-time lag on the
basis of their 14-day sample interval. We compared our LDI temperature estimates with monthly CTD
measurements between 0 and 50 m depth, the temperature at depth of maximum primary productivity
and the temperature at the chlorophyll maximum (Turich et al., 2013; http://www.imars.usf.edu/cariaco)
(Fig. 10). During the upwelling season, temperatures are significantly lower due to the upward migration
of isotherms, whereas during the non-upwelling period, temperatures are higher, particularly in the upper
20 m, and the water column is more stratified (Fig. 10). LDI underestimates SST during stratification,
which suggests that the LCD producers may thrive at depths of ca. 20–30 m. During upwelling, LDI-
temperatures agree better with SST, implying that the habitat of the LCD producers potentially was
closer to the surface, coincident with the shoaling of the nutricline and thermocline (Fig. 10). Turich et
al. (2003) found that the $U^{K'}_{37}$-derived temperatures agreed reasonably well with the measured
temperatures at the chlorophyll maximum, which is generally found below 20 m depth (average 30–34
m depth; ranging between 1 and 55 m) in the Cariaco Basin. The LDI temperatures are almost always
higher than the temperatures at the chlorophyll maximum (Fig. 10), and higher than the temperatures at
30 m depth, implying that the LDI producers may reside in the upper 30 m of the water column, which
is consistent with the results of Rampen et al. (2012), who showed that LDI-derived temperatures have



the strongest correlation with temperatures of the upper 20 m of the water column. This also agrees with
Balzano et al. (2018) who observed highest LCD abundances within the upper 20 m of the water column
in the Tropical Atlantic.
In the Mozambique Channel, the LDI temperature variations are much smaller (< 2 °C; Fig. 8a) than the
seasonal SST variation ranging between ca. 24.5 and 30.5 °C. Accordingly, during the warmest months
of the year, the difference between LDI-derived and satellite-derived SST is outside of the calibration
error (i.e., > 2 °C). However, this is similar to the $U^{K'}_{37}$ and $TEX^{H}_{86}$ which also did not reveal seasonal
variations. This lack of seasonality was explained by lateral advection and re-suspension of fine
sediment material by migrating meso-scale eddies and thus ending up in the deeply moored sediment
trap (Fallet et al., 2011; 2012). Most likely, this also explains the lack of seasonal variation in our LDI
record (Fig. 8a). Nevertheless, the average LDI temperature for the sediment trap of 26.4 °C agrees
reasonably well with the annual mean satellite-derived SST of 27.6 °C for the sampled years.
Additionally, there is a good agreement with the average LDI temperature of 26.0 °C for two underlying
surface sediments, as well as with the decadal average SST of 26.7 °C for 1955-2012 (Locarnini et al.,
2013) given by the World Ocean Atlas (2013). For the North Atlantic, we also observe rather constant
LDI temperatures during the year (Fig. 5) which contrasts with seasonal variations in satellite SSTs of
ca. 3 to 5 °C. Nevertheless, differences are mostly within the calibration error, except at M1 and M2
where during winter and spring LDI-derived temperatures are between 0.5 and 2.8 °C higher than
satellite SSTs. Similar to the LDI, also the $TEX^{H}_{86}$ and $U^{K'}_{37}$-derived SSTs for the tropical Atlantic
sediment traps do not reveal clear seasonal variation. As all three proxies show minimal seasonal
variability, this might indicate that the lipids are potentially allochtonous and partially derive from
distant regions, resulting in an integrated average temperature signal, similar to the Mozambique
Channel. Nevertheless, the flux-weighted annual LDI temperatures of the tropical Atlantic sediment
traps (26.6 for M1 and 27.1 °C for M2 and M4) agree well with the annual mean satellite-derived SSTs
of 26.1, 26.0 and 27.5 °C for M1, M2 and M4, respectively. Moreover, the LDI-derived temperatures in
the underlying sediments (26.5, 26.6 and 26.7 °C, respectively) do not only agree well with those found

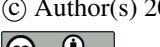


in a single year in the sediment traps but also with the decadal average SSTs for 1955 to 2012 (26.2,
27.1 and 26.3 °C, respectively; Locarnini et al., 2013; Fig. 7b).

Interestingly, $TEX^H_{86}$ temperature estimates are relatively similar for traps M2 and M4 but at M1 they
are lower than satellite SST in both the sediment trap and surface sediments (Fig. 7d). This
underestimation of SST at M1 might suggest GDGT addition from colder subsurface waters. Indeed
Balzano et al. (unpublished results) show that crenarchaeol is typically abundant between ca. 40 and
100 m water depth, agreeing with previous findings which have shown that the $TEX_{86}$ can reflect
subsurface temperatures rather than surface temperature in some regions (e.g., Huguet et al., 2007; Kim
et al., 2012; 2015; Schouten et al., 2013; Chen et al., 2014; Wuchter et al., 2006). Consequently, for the
surface sediments, we also calculated subsurface temperatures, using the calibration of Kim et al. (2012)
(Fig. 7d), and compared these with the depth-integrated annual mean temperatures of the upper 150 m
(Locarnini et al., 2013), caclulated following Kim et al. (2008), which indeed shows a better
correspondence for the eastern Atlantic surface sediment, i.e., the sediments close to M1. This is likely
caused by the steepening of the thermocline towards the east, as shown in Fig. 7a,d, in which we have
indicated the approximate production depths of the temperature proxies. The thermocline at M1 is much
steeper and shallower, which implies that GDGTs produced at ~ 100 m depth will record a lower
temperature than at M2 and M4.

**5. Conclusions**
In this study we have evaluated LCD-based proxies, particularly the LDI, in sediment trap time series
from five sites in the tropical North Atlantic, the Cariaco Basin and the Mozambique Channel. For the
North Atlantic we found that in the water column ca. 25–85 % of the export of these lipid biomarkers is
preserved during settling from 1200m to 3500m, and that generally less than 2 % was preserved in the
surface sediments. Despite substantial degradation at the seafloor, likely linked to the prolonged oxygen
exposure time, LCD-derived temperatures from the sediments are generally very similar to the annual

 

mean LCD-derived temperatures in both the deep and shallow traps as well as to annual mean SST for
the specific sampling year and on decadal time scales for the specific sites. In the Cariaco Basin we
observe a strong seasonality in the LDI which is linked to the upwelling season at temperatures
associated with a water depth of up to ca. 30 m during summer stratification, and at SST during winter
upwelling accompanied by shoaling of both the nutricline and isotherms. The LDI temperatures in the
Mozambique Channel and the tropical Atlantic reveal minimal seasonal change although seasonal SST
contrasts amount to 3-5°C. For the Mozambique Channel this is likely caused by lateral advection of re-
suspended sediment by meso-scale eddy migration, a signal not substantially altered by diagenesis.
Seasonal variations in the Diol Index are minimal in the central and western North Atlantic and 1,14-
diol concentrations are rather low, implying little *Proboscia* diatom productivity. However, in the
eastern Atlantic closest to the African continent, the Diol Index attains a clear spring maximum that is
likely associated with upwelling in the Guinea Dome during summer to autumn, suggesting the Diol
Index reflects a pre-upwelling signal, consistent with the current knowledge on *Proboscia* ecology. In
the Cariaco Basin, controlled by seasonal upwelling, the Diol Index reveals the same clear seasonal
trend observed in primary productivity, arguing that for this location the Diol Index is an excellent
indicator of upwelling intensity.

**Data availability**. The data reported in this paper is archived in PANGAEA (www.pangaea.de.)

**Author contributions**. MWdB, JSSD, and SS designed the experiments and MWdB carried them out.
JU carried out the time-series analysis. JBWS, GJAB, and RCT deployed sediment traps and collected
sediment trap materials. MWdB prepared the paper with contributions from coauthors.

**Competing interests**. The authors declare that they have no conflict of interest.




**Acknowledgements.** We are grateful to Laura Schreuder and Denise Dorhout for analytical support,
Wim Boer for help with MatLab calculations (BAYSPLINE), Laura Korte and Catarina Guerreiro for
constructive discussions, and Isla Castañeda, Ulrike Fallet and Courtney Turich for providing and
working up samples. This research has been funded by the European Research Council (ERC) under the
European Union's Seventh Framework Program (FP7/2007-2013) ERC grant agreement [339206] to
S.S. and ERC grant agreement [311152] as well as NWO project [822.01.008] to J-B.S.. S.S. and
J.S.S.D. receive financial support from the Netherlands Earth System Science Centre (NESSC) through
a gravitation grant from the Dutch ministry for Education, Culture and Science (grant number

024.002.001).

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





1171

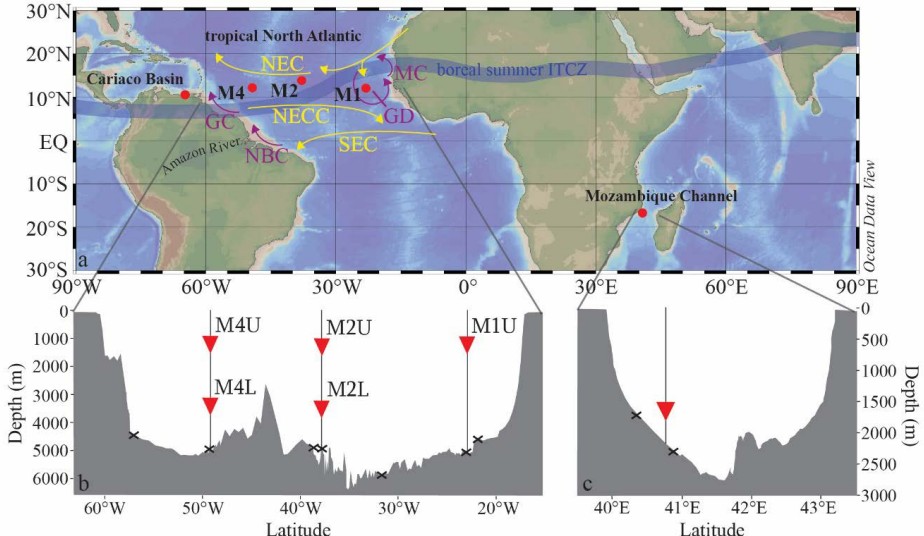

**Fig. 1** **(a)** Location map showing the five sediment trap mooring sites in the Cariaco Basin, the tropical

North Atlantic (M1, M2 and M4) and the Mozambique Channel. Two of the moorings in the tropical

North Atlantic (M2 and M4) contain an upper ('U') and a lower ('L') trap, shown in the bathymetric

section below **(b)** with traps depicted as red triangles and surface sediments shown as black crosses. A

similar section profile is shown for the Mozambique Channel (c), where also the sediment trap and the

surface sediments are indicated. All maps/sections are generated in Ocean Data View (Schlitzer, 2015).





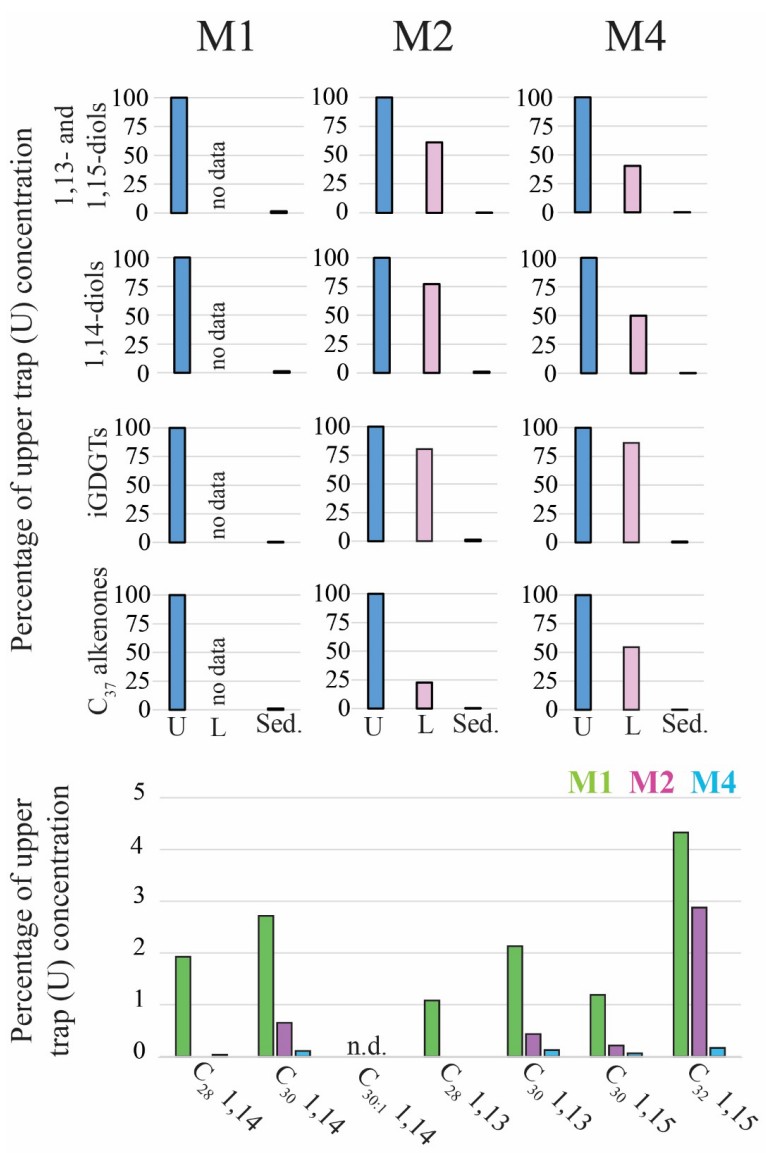

1178

**Fig. 2** Relative concentrations of biomarker lipids for the mooring sites M1, M2 and M4 in the tropical
North Atlantic. Upper panel: percentages of lipid biomarkers in the lower traps ('L'; 3500 m) and the
surface sediments ('Sed.') relative to the annual flux-weighted concentrations in the upper traps ('U';
1200 m; set at 100%). The lower panel shows the preservation of the individual LCDs (sediments versus
upper trap flux-weighted concentration) for the three sediment trap sites. For M1 and M2 the
sedimentary LCD concentration were based on the average of the two nearby underlying surface
sediments (Fig. 1). When no bar is shown than the LCD was not detected.

1186





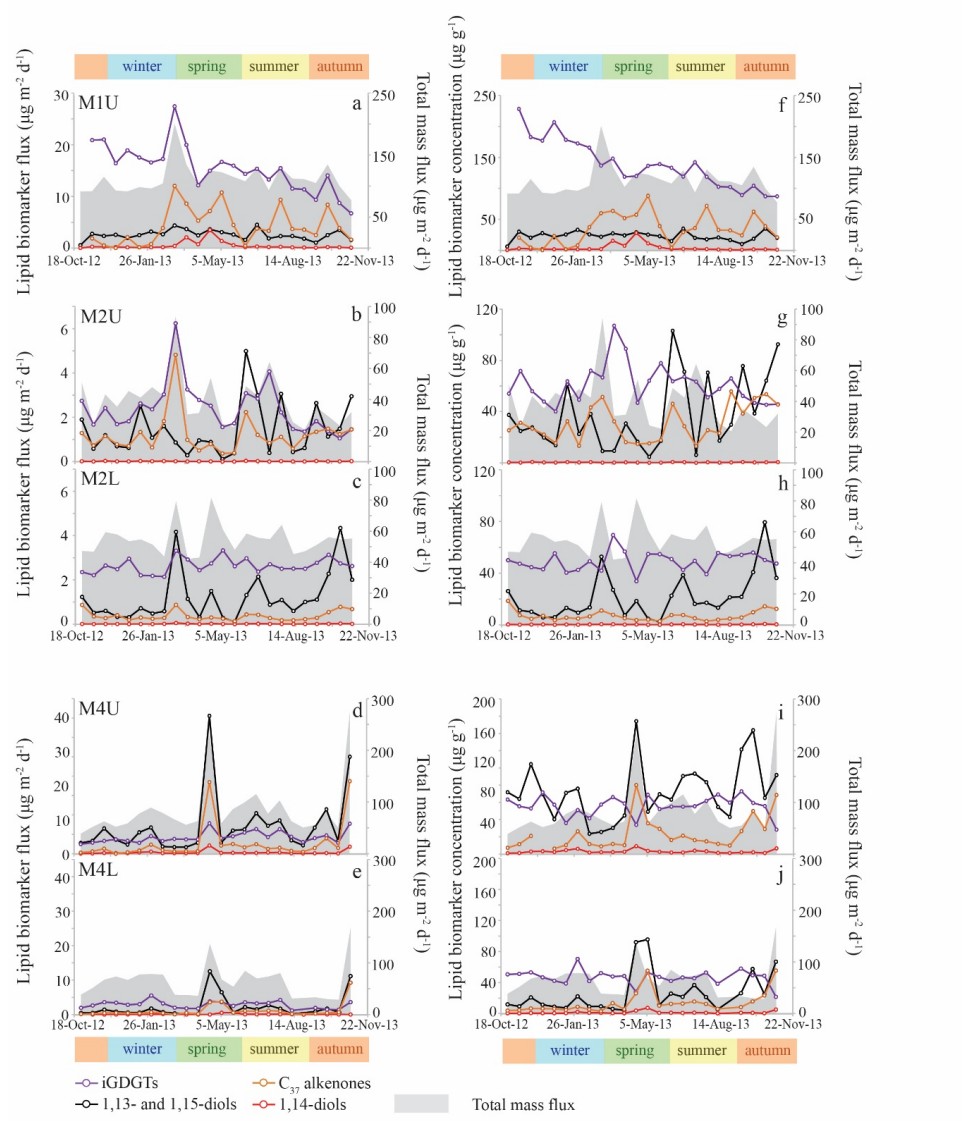

**Fig. 3** Lipid biomarker fluxes for the tropical North Atlantic sediment traps, i.e., M1, upper and lower M2, and upper and lower M4 in panels **(a)** to **(e)**. Lipid biomarker fluxes (iGDGTs in purple; $C_{37}$ alkenones in orange; 1,13- and 1,15-diols in black; 1,14-diols in red) are indicated on the left *y*-axis, and the total mass flux (grey stack; Korte et al., 2017) on the right *y*-axis. Lipid biomarker concentrations are plotted in panels **(f)** to **(j)**, with biomarker concentrations on the left *y*-axis, and the total mass flux on the right *y*-axis. Note that the *y*-axes are different per sediment trap site, but identical for upper (U) and lower (L) traps.





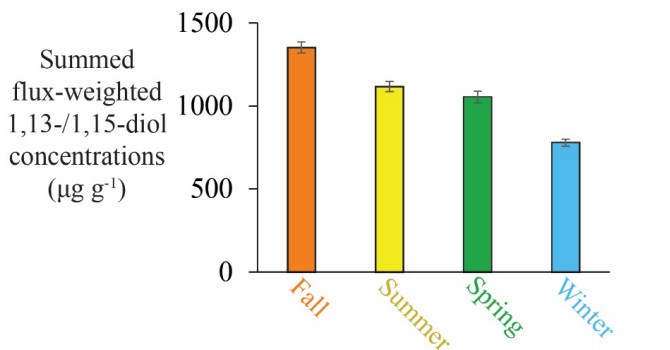


**Fig. 4** Seasonal summed flux-weighted average of 1,13-/1,15-diol concentrations in all sediment traps

(station M1 upper trap, station M2 upper and lower trap and station M4 upper and lower trap) of the

tropical North Atlantic.




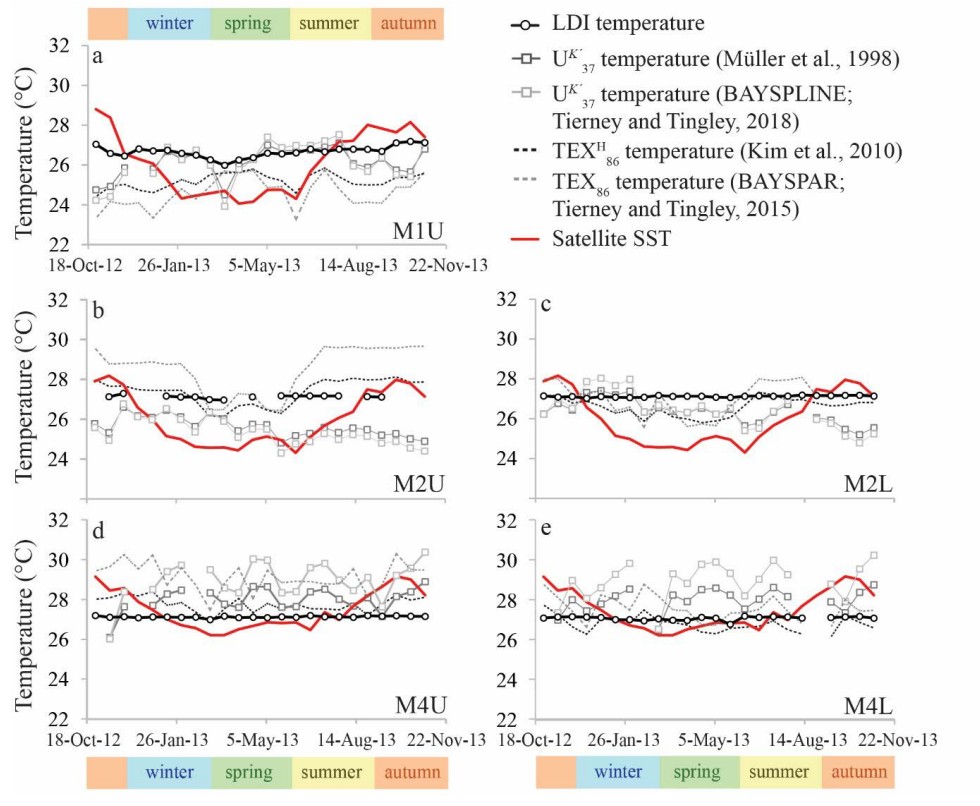


**Fig. 5** Temperature proxy records for the tropical North Atlantic. Panel **(a)** shows upper trap station
M1, **(b)** upper trap station M2 and **(c)** lower trap M2, respectively, **(d)** upper trap station M4 and **(e)**
lower trap station M4, respectively.






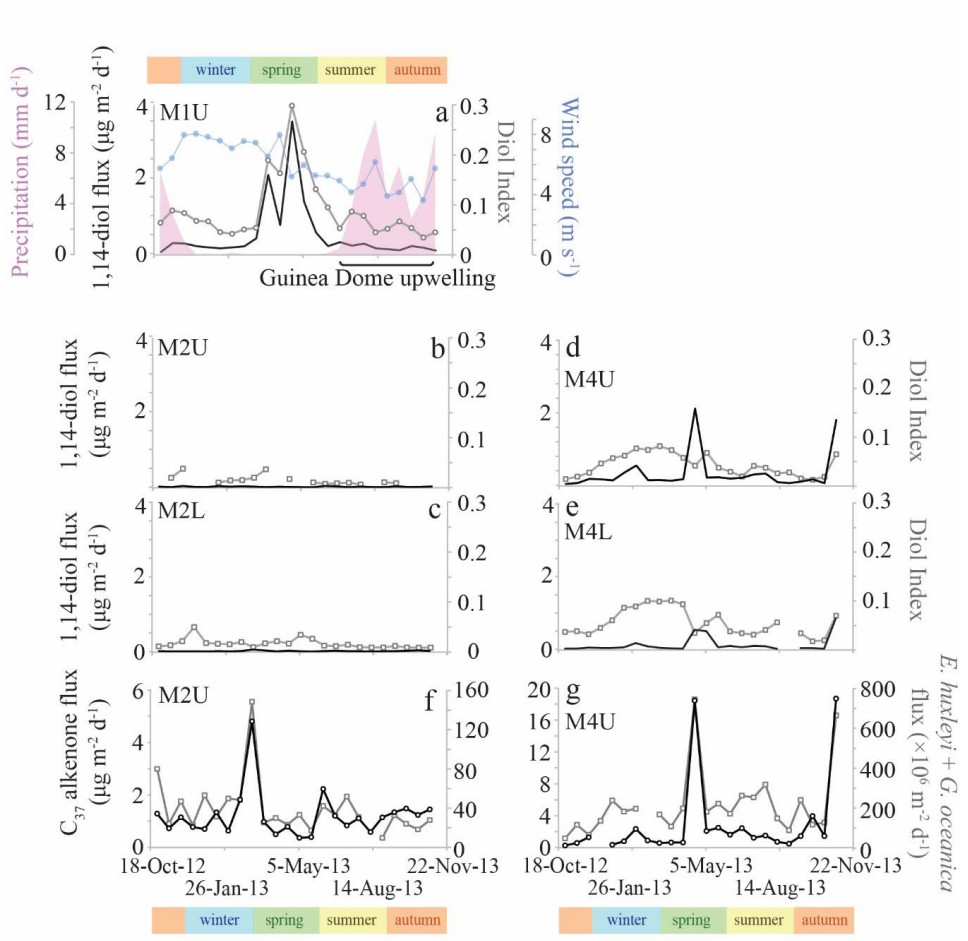

**Fig. 6** Phytoplankton productivity records for the tropical North Atlantic. Panels **(a)** – **(e)** show the 1,14-
diol fluxes (left $y$-axis; black) and the Diol Index (right $y$-axis; grey) for sediment traps. The $y$-axes are
the same for these panels. Wind speed and precipitation data were adapted from Guerreiro et al. (in
revision); for references regarding remote sensing parameters, see Guerreiro et al. (2017). Panels **(f)** and
**(g)** show the $C_{37}$ alkenone fluxes (left $y$-axis; black) and combined fluxes of *E. huxleyi* and *G. oceanica*
(from Guerreiro et al., 2017; right $y$-axis; grey) for the upper traps of M2 and M4.

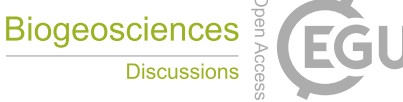



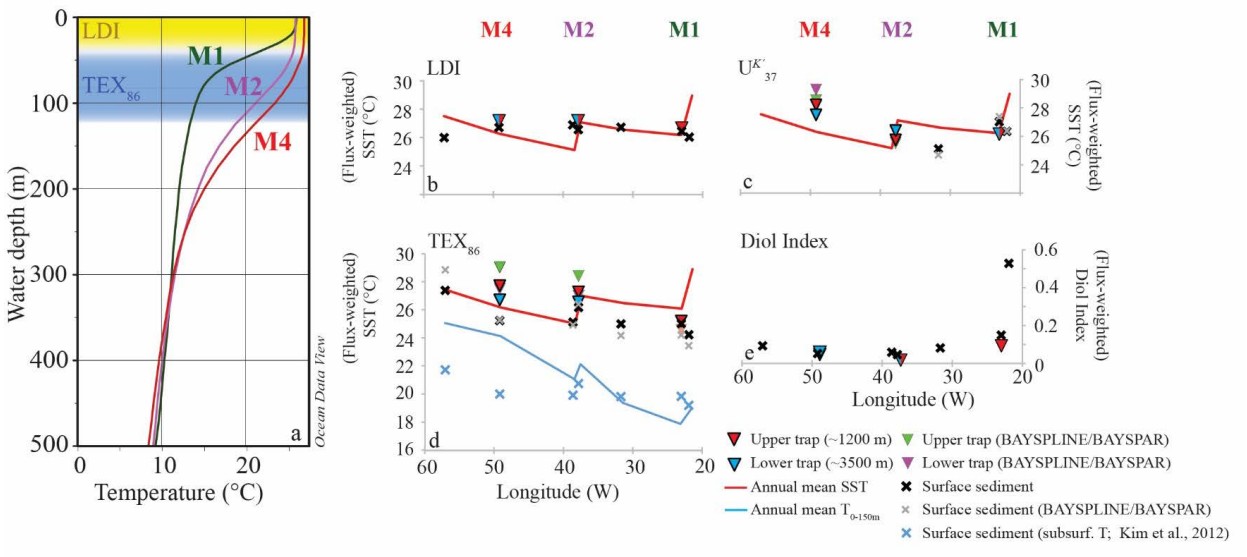


**Fig. 7 (a)** Annual mean temperature profiles at the sediment trap locations (World Ocean Atlas 2013)
with approximate proxy-lipid production depths indicated, as deduced from Balzano et al. (unpublished
results). **(b)** Flux-weighted average (annual) proxy results for the sediment traps compared with the
underlying sediments (crosses) and annual mean SST (red line; World Ocean Atlas 2013). Panel **(b)**, **(c)**
and **(d)** show the LDI, $U^{K'}_{37}$ and $TEX_{86}$ temperature results, respectively. Triangles reflect sediment trap
results (red = upper/~1200 m; blue = lower/~3500 m), and crosses represent surface sediments. In case
of the $U^{K'}_{37}$ and $TEX_{86}$, the green and purple triangles and grey crosses reflect the temperatures
calculated using the BAYSPLINE and BAYSPAR models (Tierney and Tingley, 2014; 2015; 2018),
whereas the other temperatures were calculated by means of the Müller et al. (1998) and Kim et al.
(2010; $TEX^{H}_{86}$) calibrations, respectively. The blue line and crosses in panel **(d)** reflect the depth-
integrated temperature for the upper 0-150 m, and subsurface $TEX^{H}_{86}$ temperatures (Kim et al., 2012).
Panel **(e)** shows the flux-weighted average Diol Index values for the sediment traps, and the Diol Index
estimates for the surface sediments.





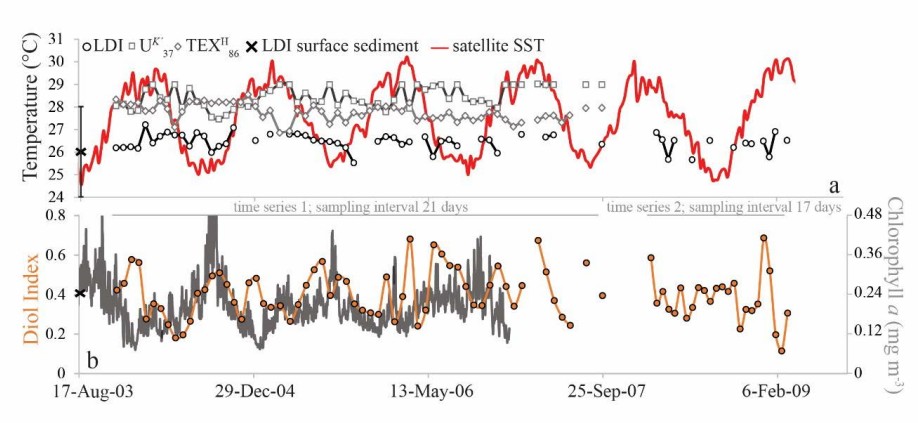

1226

**Fig. 8** The LDI-derived temperatures, together with the TEX$^H_{86}$ and U$^{K'}_{37}$-derived temperatures and
satellite SST (Fallet et al., 2011) (**a**) and the Diol Index (**b**) for the Mozambique Channel sediment trap.
The black cross in panel (**a**) reflects the average LDI temperature of two underlying surface sediments,
with the LDI calibration error. The chlorophyll *a* data is from Fallet et al. (2011).






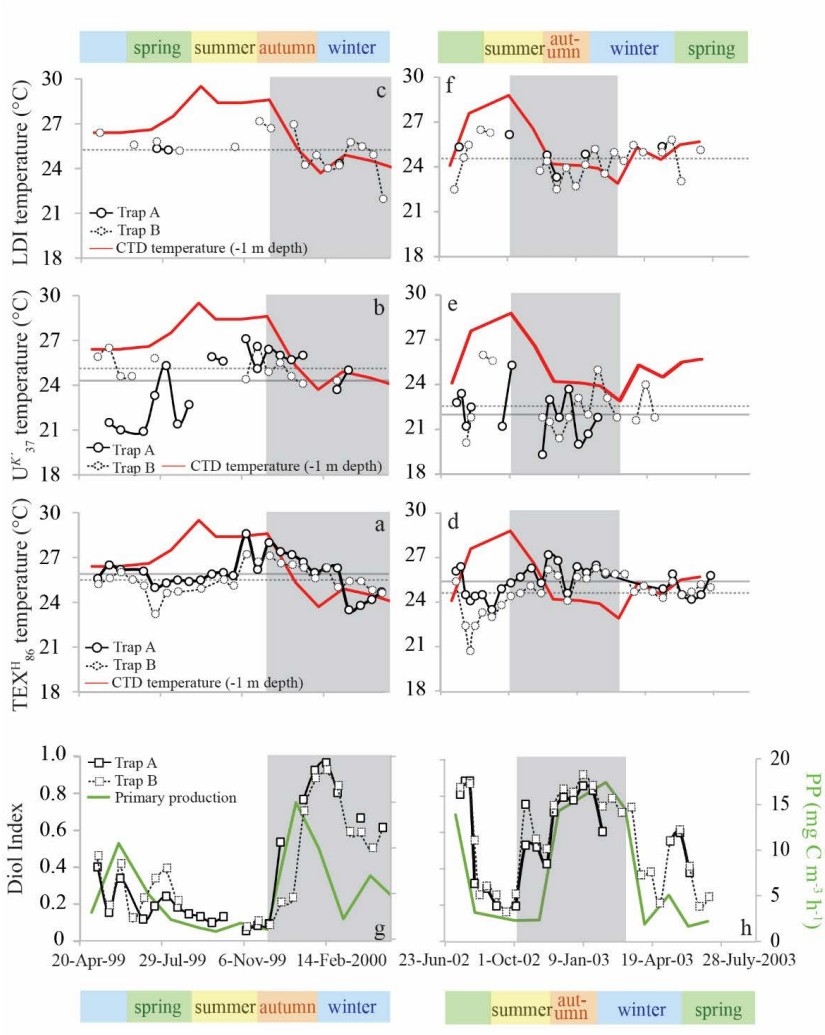

**Fig. 9** Seasonal proxy derived temperature and upwelling/productivity records for the sediment traps in
the Cariaco Basin. Panels **(a)**, **(b)** and **(c)** show the May 1999 – May 2000 time series $TEX^H_{86}$-, $U^{K'}_{37}$-
and LDI-derived temperature reconstructions for Trap A (275 m depth; solid symbols) and Trap B (455
m depth; dashed symbols), respectively. Panels **(d)**, **(e)** and **(f)** show the proxy data for the July 2002 –
July 2003 time series, with CTD-temperatures (1 m depth) in red. The $U^{K'}_{37}$, $TEX^H_{86}$ and CTD
temperatures are adopted from Turich et al. (2013). The horizontal lines reflect the average proxy-
derived temperatures (Trap A = solid; Trap B = dashed). Panel **(g)** and **(h)** show the 1,14-diol based
Diol Index (Rampen et al., 2008) for the 1999-2000 and 2002-2003 time series, respectively, for Trap
A (275 m depth; solid symbols) and Trap B (455 m depth; dashed symbols). Primary productivity in mg
C m$^{-3}$ h$^{-1}$ is plotted in green (data adopted from Turich et al., 2013). The shaded area reflects the period
of upwelling.



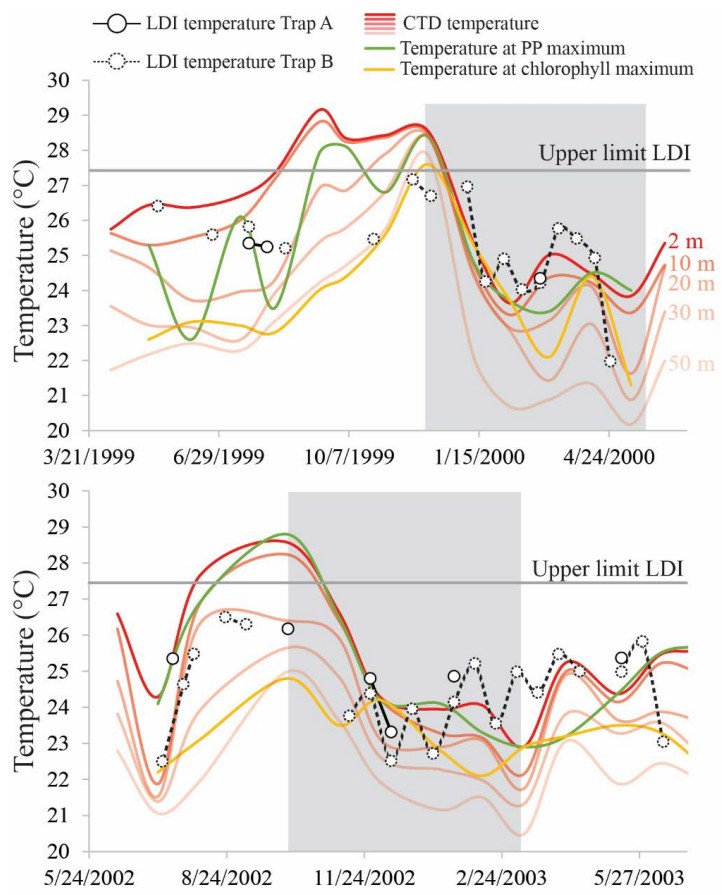

**Fig. 10** LDI temperature records for the Cariaco Basin time series May 1991 – May 2000 and July 2002 – July 2003 for Trap A (275 m depth; solid symbols) and Trap B (455 m depth; dashed symbols), with CTD-derived temperatures at 2, 10, 20, 30 and 50 m depth (in red; http://www.imars.usf.edu/CAR/index.html; CARIACO time series composite CTD profiles), the temperature at the depth of maximum primary production (green) and the temperature at the depth of the chlorophyll maximum (yellow; data adapted from Turich et al., 2013). The shaded area represents the upwelling season.