# Peer review of "Long chain diols in settling particles in tropical oceans: insights into sources, seasonality and proxies."

_Biogeosciences, 2019_

## Referee Comment (RC1) · Anonymous Referee #1 · 11 Feb 2019

**Review on bg-2019-15 "Long chain diols in settling particles in tropical oceans: insights into sources, seasonality and proxies" by de Bar et al.**

This manuscript investigates long-chain diols (LCDs) in sediment trap time series from five tropical sites (tropical North Atlantic, Cariaco Basin, Mozambique Channel) to assess seasonal variations in fluxes of LCDs and associated proxies (Long chain Diol Index and Diol Index). These data are compared with other lipid proxies (alkenones and GDGTs) and previous published data (primary production, SST,..). Results show that surface sediment LDI temperatures in the Atlantic and Mozambique Channel compare well with the average LDI-derived temperatures from the overlying sediment traps, as well as with decadal annual mean SST. In the Mozambique Channel and the tropical Atlantic, the LDI temperatures reveal minimal seasonal change although there are clear seasonal SST contrasts, which is likely due to lateral advection of re-suspended sediment. In the Cariaco Basin, a strong seasonality in the LDI is observed, which is linked to the upwelling season and stratification of the water column. In addition, in the Atlantic, the Diol Index reflects a pre-upwelling signal, whereas in the Cariaco Basin, the Diol Index seems to be an indicator of upwelling intensity.

This paper is a valuable contribution to the understanding of the seasonal production of LCDs in marine environments and how it is translated in the temperature proxy LDI and the Diol Index (upwelling proxy). A strength of the paper is that the LCD data has been compared with other available data for each site (primary production, SST, alkenones, GDGTs,…), which gives a broader picture and supports the interpretations based on LCDs. The writing style is clear and precise and the interpretations are generally supported by the data. This manuscript is thus suitable for Biogeosciences. However, the current manuscript could be improved before publication. Please find my comments below.

**General comments:**

Diol index and upwelling: The authors argue that, in the Cariaco Basin, the Diol Index is an excellent indicator of upwelling intensity (Lines 476-480). However, when looking at the 1999-2000 time series, high values of the diol index actually occur when the primary production decreases. What are the $R^2$ values (and *p* values) that justify "a strong correlation with primary production rates"?

In addition, for the eastern Atlantic (M1 trap), the authors argue that the Diol Index reflects a pre-upwelling signal, consistent with the current knowledge on *Proboscia* ecology (Lines 509-526). I would like to see more discussion that explains why at one location the Diol index indicates pre-upwelling conditions, whereas it seems to be an indicator of upwelling intensity at another location.

Keto-ols as oxidation products (Lines 578-586): An alternative explanation for the non-detection of 1,14-keto-ols would be that keto-ols are not oxidation products of LCDs, but rather produced by unknown organism(s) (Versteegh et al., 1997). Previous studies have indeed shown the absence of evidence of conversion of diols into their corresponding oxidized keto-ols (Jiang et al., 1994; Méjanelle al 2003; Shimokawara et al., 2010). I think the authors should acknowledge this.

*References*:

Jiang, S.C., O'Leary, T., Volkman, J.K., Zhang, H.Z., Jia, R.F., Yu, S.H., Wang, Y., Luan, Z.F., Sun, Z.Q., Jiang, R.H., 1994. Origins and simulated thermal alteration of sterols and keto-alcohols in deep-sea marine sediments of the Okinawa Trough. Organic Geochemistry 21, 415-433.

Méjanelle, L., Sanchez-Gargallo, A., Bentaleb, I., Grimalt, J.O., 2003. Long chain *n*-alkyl diols, hydroxyl ketones and sterols in a marine eustigmatophyte, *Nannochloropsis gaditana*, and in *Brachionus plicatilis* feeding on the algae. Organic Geochemistry 34, 527-538.

Shimokawara, M., Nishimura, M., Matsuda, T., Akiyama, N., Kawai, T. 2010. Bound forms, compositional features, major sources and diagenesis of long chain, alkyl mid-chain diols in Lake Baikal sediments over the past 28,000 years. Organic Geochemistry 41, 753-766.

Versteegh, G.J.M., Bosch, H.J., de Leeuw, J.W., 1997. Potential palaeoenvironmental information of $C_{24}$ to $C_{30}$ mid-chain diols, keto-ols and mid-chain hydroxy fatty acids; a critical review. Organic Geochemistry 27, 1-13.

Figures: I think the current order of the figures does not necessarily follow the logic of the results/discussion. For more clarity, I would suggest modifying the order as follows: Fig. 2 should be Fig. 9; Fig. 3 should be Fig. 2; Fig. 4 should be Fig. 8; Fig. 5 should be Fig. 3; Fig. 6 should be Fig. 4; Fig. 7 should be Fig. 5; Fig. 8 should be Fig. 6; Fig. 9 should be Fig. 7.

**Specific comments:**

Line 25: specify "with emphasis on the temperature proxy Long Chain Diol Index".

Line 27: specify "similar to the two other lipid-based temperature proxies $TEX_{86}$ and $U^{K'}_{37}$".

Line 27: "In addition" instead of "However".

Line 29: Could be rephrased as: "In contrast, the LDI in the Cariaco Basin shows larger seasonal variation".

Line 48: Need references.

Lines 48-50: Could be rephrased as: "However, research showed that despite their highest abundance being recorded in the upper 100 m of the water column, Thaumarchaeota can be present down to 5000 m depth (Karner et al., 2001; Herndl et al., 2005)".

Line 69: "for autumn to summer" should be "for autumn and summer".

Figure 1: indicate in the caption what NEC, NECC, SEC, MC, GD, NBC and GC stand for. Is it possible to add the position of the ITCZ during the boreal winter?

Line 200: What are CTD measurements?

Line 256-258: Could diols be found in the DCM:MeOH (1:1; v/v) fraction? Have you checked?

Line 369: Should be as: "$C_{28}$ and $C_{30}$ 1,13- (0–3 %), the $C_{30}$ 1,15- (44–99 %), and the $C_{32}$ 1,15-diols (0–7 %)".

Lines 367-376: I think a table showing the presence/absence for each diols (and the % of total LCDs) at the different traps (M1, M2,...) and different sites (Atlantic, Mozambique Channel, Cariaco Basin) would be useful to clearly see which diols are detected for each location. The Figure 2 is used to discuss the preservation between traps and sediments rather than showing the diols detected.

Line 392: Fig. 5 is cited before Fig. 4. I think the order of the figures should be changed (see previous comment).

Line 397: cite Figure 7.

---

## Author Comment (AC1) · 13 Feb 2019

**Response to referee #1**

Review: This manuscript investigates long-chain diols (LCDs) in sediment trap time series from five tropical sites (tropical North Atlantic, Cariaco Basin, Mozambique Channel) to assess seasonal variations in fluxes of LCDs and associated proxies (Long chain Diol Index and Diol Index). These data are compared with other lipid proxies (alkenones and GDGTs) and previous published data (primary production, SST,..). Results show that surface sediment LDI temperatures in the Atlantic and Mozambique Channel compare well with the average LDI-derived temperatures from the overlying sediment traps, as well as with decadal annual mean SST. In the Mozambique Channel and the tropical Atlantic, the LDI temperatures reveal minimal seasonal change although there are clear seasonal SST contrasts, which is likely due to lateral advection of re-suspended sediment. In the Cariaco Basin, a strong seasonality in the LDI is observed, which is linked to the upwelling season and stratification of the water column. In addition, in the Atlantic, the Diol Index reflects a pre-upwelling signal, whereas in the Cariaco Basin, the Diol Index seems to be an indicator of upwelling intensity. This paper is a valuable contribution to the understanding of the seasonal production of LCDs in marine environments and how it is translated in the temperature proxy LDI and the Diol Index (upwelling proxy). A strength of the paper is that the LCD data has been compared with other available data for each site (primary production, SST, alkenones, GDGTs,…), which gives a broader picture and supports the interpretations based on LCDs. The writing style is clear and precise and the interpretations are generally supported by the data. This manuscript is thus suitable for Biogeosciences. However, the current manuscript could be improved before publication. Please find my comments below.

**We thank the referee for the positive assessment and for the comments, which we will discuss below.**

General comments:

Diol index and upwelling: The authors argue that, in the Cariaco Basin, the Diol Index is an excellent indicator of upwelling intensity (Lines 476-480). However, when looking at the 1999-2000 time series, high values of the diol index actually occur when the primary production decreases. What are the $R^2$ values (and p values) that justify "a strong correlation with primary production rates"?

**We agree with the reviewer that for the 1999-2000 time series there is a disagreement during January/February when the diol index increases and primary production rates decrease. We will mention this in the revised version of the manuscript. The 'strong correlation' between the diol index and primary production is based purely on the visual agreement between both time series. We were not able to perform a correlation analysis since the data are differently spaced in time. We will also emphasize this in the revised version manuscript.**

In addition, for the eastern Atlantic (M1 trap), the authors argue that the Diol Index reflects a preupwelling signal, consistent with the current knowledge on Proboscia ecology (Lines 509-526). I would like to see more discussion that explains why at one location the Diol index indicates preupwelling conditions, whereas it seems to be an indicator of upwelling intensity at another location.

**We agree that this seems contradictory and requires more discussion, which we will implement in the revised version of the manuscript. The Diol Index is an upwelling indicator based on the assumption that *Proboscia* diatoms generally thrive in upwelling regions. However, the index is in fact an indicator for *Proboscia* productivity, and whether it reflects upwelling/pre-upwelling/stratification/etc. conditions will depend on the region and the local ecological dynamics determining the role of *Proboscia* diatoms (e.g., Rampen et al., 2014; de Bar et al., 2018). Studies**

**have shown that *Proboscia* diatoms are often more dominant during early/pre-upwelling because they need relatively little silica and they are able to migrate to deeper waters to obtain nutrients (Koning et al., 2001) and sediment trap data from Wakeham et al. (2002), Prahl et al. (2000), Sinninghe Damsté et al. (2003) and Rampen et al. (2007) show that *Proboscia* lipids (diols and/or hydroxyl methyl alkanoates) are highest during early upwelling. Therefore, we hypothesize that this Diol Index maximum during spring which we observe for station M1 in the Atlantic might be a pre/early-upwelling signal since the upwelling in the Guinea Dome often occurs between July and October (Siedler et al., 1992). Indeed, *Proboscia* diatoms do not reflect early-upwelling in every region. Reports of *Proboscia* spp. blooms vary from stratification to early-upwelling to postbloom, and from high nutrients to low nutrients (see Rampen et al., 2014; references in Table 1). Apparently, in the Cariaco Basin, *Proboscia* diatoms bloom relatively synchronous with general productivity, as evidenced from the agreement between the Diol Index and primary production time series, emphasizing the value of sediment trap studies like this in revealing regional differences in proxy signals.**

Keto-ols as oxidation products (Lines 578-586): An alternative explanation for the non-detection of 1,14-keto-ols would be that keto-ols are not oxidation products of LCDs, but rather produced by unknown organism(s) (Versteegh et al., 1997). Previous studies have indeed shown the absence of evidence of conversion of diols into their corresponding oxidized keto-ols (Jiang et al., 1994; Méjanelle al 2003; Shimokawara et al., 2010). I think the authors should acknowledge this.

**We agree, and we will mention this hypothesis as well.**

References:
Jiang, S.C., O'Leary, T., Volkman, J.K., Zhang, H.Z., Jia, R.F., Yu, S.H., Wang, Y., Luan, Z.F., Sun, Z.Q., Jiang, R.H., 1994. Origins and simulated thermal alteration of sterols and keto-alcohols in deepsea marine sediments of the Okinawa Trough. Organic Geochemistry 21, 415-433.
Méjanelle, L., Sanchez-Gargallo, A., Bentaleb, I., Grimalt, J.O., 2003. Long chain n-alkyl diols, hydroxyl ketones and sterols in a marine eustigmatophyte, Nannochloropsis gaditana, and in Brachionus plicatilis feeding on the algae. Organic Geochemistry 34, 527-538.
Shimokawara, M., Nishimura, M., Matsuda, T., Akiyama, N., Kawai, T. 2010. Bound forms, compositional features, major sources and diagenesis of long chain, alkyl mid-chain diols in Lake Baikal sediments over the past 28,000 years. Organic Geochemistry 41, 753-766.
Versteegh, G.J.M., Bosch, H.J., de Leeuw, J.W., 1997. Potential palaeoenvironmental information of C24 to C30 mid-chain diols, keto-ols and mid-chain hydroxy fatty acids; a critical review. Organic Geochemistry 27, 1-13.

Figures: I think the current order of the figures does not necessarily follow the logic of the results/discussion. For more clarity, I would suggest modifying the order as follows: Fig. 2 should be Fig. 9; Fig. 3 should be Fig. 2; Fig. 4 should be Fig. 8; Fig. 5 should be Fig. 3; Fig. 6 should be Fig. 4; Fig. 7 should be Fig. 5; Fig. 8 should be Fig. 6; Fig. 9 should be Fig. 7.

**We will reconsider the order of the figures.**

Specific comments:

Line 25: specify "with emphasis on the temperature proxy Long Chain Diol Index".

**We will adjust this.**

Line 27: specify "similar to the two other lipid-based temperature proxies TEX86 and UK'37".

**We will adjust this.**

Line 27: "In addition" instead of "However".

**We will adjust this.**

Line 29: Could be rephrased as: "In contrast, the LDI in the Cariaco Basin shows larger seasonal variation".

**We will adjust this.**

Line 48: Need references.

**We will add references.**

Lines 48-50: Could be rephrased as: "However, research showed that despite their highest abundance being recorded in the upper 100 m of the water column, Thaumarchaeota can be present down to 5000 m depth (Karner et al., 2001; Herndl et al., 2005)".

**We will adjust this.**

Line 69: "for autumn to summer" should be "for autumn and summer".

**We will adjust this.**

Figure 1: indicate in the caption what NEC, NECC, SEC, MC, GD, NBC and GC stand for. Is it possible to add the position of the ITCZ during the boreal winter?

**We will clarify the abbreviations and indicate the position of the ITCZ during boreal winter.**

Line 200: What are CTD measurements?

**We refer here to temperature measurements of seawater at 1m water depth sampled by CTD. We will clarify this.**

Line 256-258: Could diols be found in the DCM:MeOH (1:1; v/v) fraction? Have you checked?

**We thank the reviewer for noticing this, since the sentence is incorrect: not the MeOH fractions were analyzed for diols, but the DCM:MeOH (1:1) fractions. We will correct this.**

Line 369: Should be as: "C28 and C30 1,13- (0–3 %), the C30 1,15- (44–99 %), and the C32 1,15-diols (0–7%)".

**We will adjust this.**

Lines 367-376: I think a table showing the presence/absence for each diols (and the % of total LCDs) at the different traps (M1, M2,...) and different sites (Atlantic, Mozambique Channel, Cariaco Basin) would be useful to clearly see which diols are detected for each location. The Figure 2 is used to discuss the preservation between traps and sediments rather than showing the diols detected.

**We are not sure whether we agree, since the number of figures is already extensive, as is the result section, and we consider this relatively detailed.**

Line 392: Fig. 5 is cited before Fig. 4. I think the order of the figures should be changed (see previous comment).

**We will adjust the order of the figures.**

Line 397: cite Figure 7.

**We will cite Fig. 7 here.**

---

## Referee Comment (RC2) · Anonymous Referee #2 · 22 Feb 2019

General comments

In this study, de Bar et al. presented long-chain diol (LCD) data from five sites; three along a longitudinal transect in the tropical Atlantic, the Cariaco Basin and the Mozambique Channel. LCD derived indices, i.e. Long-chain Diol Index (LDI) and Diol Index, are used to reconstruct past SST and upwelling, respectively. These proxies are relatively new compared to those based on alkenones and GDGTs, thus have not been as well studied. This is where this study comes in. de Bar et al analyzed LCDs from sediment traps and underlying sediments. For the sites where alkenones and GDGT data do not yet exist, the authors also analyzed these biomarkers in addition to LCDs

to allow multi-proxy comparison for all the sites. The well-designed experiment thus allows the authors to investigate various aspects of the LCDs and their associated proxies, including the temporal evolution (seasons to years), settling processes, as well as comparison with other commonly applied biomarker proxies. The data presented by de Bar et al. generally show that LDI-derived temperatures agree within error with instrumental data in the Atlantic, albeit with different amplitude of change. At upwelling sites, the Diol Index seems to either record a pre-upwelling signal or show the same trend as in primary productivity.

The study fits the scope of Biogeosciences, and will also be of interest to readers from other community such as paleoclimate. The manuscript is generally well-written and accessible. I do, however, feel that some figures could be further improved for clarity. I find the "Results" section too long and some discussion unclear or not fully supported by the data, especially in section 4.3. Below are suggestions and comments that I hope will help the authors in further improving the manuscript. Once the concerns are addressed, I strongly recommend the publication of this manuscript.

Specific comments

**Line 34-36: Clunky sentence. Please rephrase.**

**Line 43: "Conte 2006" should be "Conte et al 2006"**

**Line 96-97: "ITCZ migrates southward during boreal winter" - would be useful to have this marked in Figure 1 too.**

**Line 100: Insert abbreviation (SEC) after South Equatorial Current.**

**Line 116: replace "/" with either a space or comma.**

**Line 119: "as result" should be "as a result"**

**Line 125: "latitudinal transect" is a transect across latitudes. What you have is a "longitudinal transect", i.e. with sites spanning longitudes at a fixed latitude (∼12°N).**

**Line 183-184: Varved sediments have annual resolution. Since you mentioned "annually to decadally resolved climate records", do you mean "laminated sediments" instead?**

**Line 224: "weight sub-aliquots" is confusing. Suggested rephrasing "sub-aliquots (by weight)".**

**Line 237-238: Confusing sentence. Sounds like you analyzed both ketone and GDGT fractions by both GC and GC/MS - which is likely not the case. Please rephrase.**

**Line 285-287: Technically this is a variant of the original BIT index proposed by Hopmans et al 2004. Please rephrase the paragraph to reflect this.**

**Line 296: This is not the first time GC is mentioned in the manuscript. Spell out "gas chromatograph" at the first mention instead of here. Also, there is no need to define the abbreviation at each mention.**

**Line 308-309: Tierney and Tingley (2018) is not the first to notice the warm-end limit of UK'37, i.e. an issue which has been in debate since the 90s. Please include the original references.**

**Line 313: "gas chromatograph (GC)" see comment to Line 296.**

**Line 314: "mass spectrometer (MS)" see comment to Line 296.**

**Section 2.5 Time-series analysis: Since the result of the time-series analysis is not a main part of the results and discussion, I would suggest to either (A) remove this rather long section or (b) move it to the supplement and add a supplementary figure depicting the result (which is briefly discussed in the text but not shown).**

**Section 3. Results: I had a hard time going through the 4-page long results section. Given the large data set spanning several sites and including several biomarkers and their associated proxies (for which I applaud the authors), this is perhaps inevitable. But I think that it will make the section more accessible for the reader if the authors**

could reduce the text by 10 to 20%, either by restructuring the text, tabulating some of the results and/or limiting the result description to only the main findings that are discussed in the following section.

**Line 362: "longitudinal" not "latitudinal".**

**Line 368-369: Confusing. Rephrase please.**

**Line 430-431: "during January and July" - replace with "between January and July". Also, it is not clear at all in Fig 5d that the TEX86H temperatures are lower during these months. Please rephrase.**

**Line 444: I'd argue that there's some structural similarity between the Diol Index and chlorophyll-a records.**

**Line 482: What are "15 and 18°"? Latitude?**

**Line 491-497: I strongly urge the authors to at least show the wavelet analysis in the supplementary info to support their claim. Please also mark the cool water events in Figure 8b to support the claim that "...the timing of the observed time periods of enhanced Diol Index variability are similar to those of the cool water events..."**

**Line 496-497: I am not following this. Assuming a sampling interval of 21 days - that would give us about 21 data points per year. With so few data points in the time series, it would be impossible to detect 4 cycles in the first half of 2006. Please clarify.**

**Line 498-499: It would be helpful to mark the timing of the passage of eddies in Fig 8b.**

**Line 504: "Fig 5c"shows LDI not Diol Index.**

**Line 508: "Fig 5e" shows LDI not Diol Index.**

**Line 522: Change "due its closer vicinity" to "due to its closer vicinity"**

**Line 523: "NW Africa" This is mentioned only once in the text. Spell out NW.**

**Line 556: r (and p) values are more appropriate as a metric to describe the correlation between two variables than r2 (which is used to describe how well the data fit the linear regression model).**

**Line 570-571: Explain briefly why one can expect LCD and levoglucosan to have similar response to degradation, e.g. in terms of their chemical behavior/structure.**

**Line 578: "for" or "in" the Atlantic?**

**Line 583-586: Include in the sentence the producers of 1,13- and 1,15-diols.**

**Line 614: Replace "minimal differences" with "minimal variations/variability".**

**Line 625-627: It is true for LDI and UK'37 that the difference between proxy temperatures and instrumental SST increase during the warmer months, but not for TEX86H. The difference between TEX86H and SST for the cooler months are almost as large as that during the warmer months, and these differences are within the calibration error. Please rephrase the sentence to reflect this.**

**Line 638-640: Taken into account proxy uncertainty, I do not think it is possible to discern if the LDI temperatures are closer to SST or 20m (some temperatures are even higher than SST!), as the isotherms of the upper 30m are so close to each other anyway during the upwelling season. In any case, a habitat depth of the upper 20m is consistent with previous studies as well (as mentioned in line 646 - 649). Please rephrase the sentence.**

**Line 676-690: This discussion is not supported by the <2°C of temperature difference between TEX86H and satellite-SST that is well within the calibration error of TEX86H. In fact, the difference is even smaller than that between the LDI temperature and satellite SST in the North Atlantic (Fig 5), which the authors did not discuss since the differences are mostly within the calibration error. The authors also need to justify why they compared the 0-150m (instead of from the same water depths as the calibration) temperatures with the temperature estimates calculated using the 0-200m calibration.**

Since the focus of the paper is on LCD proxies, and this subsurface TEX86 finding was not mentioned in the abstract nor the conclusions, I would suggest to remove this paragraph.

**Line 700-703: See comment on #Line 638-640.**

**Fig. 2: It took me a while to understand this figure. I think stacked bar chart would make a better option here, so instead of 12 panels with 3 bars each, you'd have 12 stacked bars which give you the same amount of information.**

**Line 1184: Change "concentration" to "concentrations".**

**Line 1185: Change "than" to "then".**

**Fig. 3: It is impossible to tell which lines/variables correspond to which y-axes without going through the caption. I would suggest to change the color of the right y-axis and its label (Total mass flux) to grey, i.e. the same color as the plot for the variable.**

**Fig. 4: This figure is mentioned for the first time at line 5XX in the section "Discussion" - I suggest to renumber it according to the order of its appearance in the text.**

**Fig. 7: Specify at least in the caption if the annual mean WOA SST is averaged over latitudes or at a fixed latitude. I would also remove the panel on the left and the annual mean T0-150m in panel d if line 676-690 are removed.**

---

## Author Comment (AC2) · 5 Mar 2019

**Response to referee #2**

General comments

In this study, de Bar et al. presented long-chain diol (LCD) data from five sites; three along a longitudinal transect in the tropical Atlantic, the Cariaco Basin and the Mozambique Channel. LCD derived indices, i.e. Long-chain Diol Index (LDI) and Diol Index, are used to reconstruct past SST and upwelling, respectively. These proxies are relatively new compared to those based on alkenones and GDGTs, thus have not been as well studied. This is where this study comes in. de Bar et al analyzed LCDs from sediment traps and underlying sediments. For the sites where alkenones and GDGT data do not yet exist, the authors also analyzed these biomarkers in addition to LCDs C1 to allow multi-proxy comparison for all the sites. The well-designed experiment thus allows the authors to investigate various aspects of the LCDs and their associated proxies, including the temporal evolution (seasons to years), settling processes, as well as comparison with other commonly applied biomarker proxies. The data presented by de Bar et al. generally show that LDI-derived temperatures agree within error with instrumental data in the Atlantic, albeit with different amplitude of change. At upwelling sites, the Diol Index seems to either record a pre-upwelling signal or show the same trend as in primary productivity.

The study fits the scope of Biogeosciences, and will also be of interest to readers from other community such as paleoclimate. The manuscript is generally well-written and accessible. I do, however, feel that some figures could be further improved for clarity. I find the "Results" section too long and some discussion unclear or not fully supported by the data, especially in section 4.3. Below are suggestions and comments that I hope will help the authors in further improving the manuscript. Once the concerns are addressed, I strongly recommend the publication of this manuscript.

**We thank the referee for the positive assessment and for the comments, which we will discuss below.**

Specific comments

**Line 34-36: Clunky sentence. Please rephrase.**

**We will rephrase this sentence.**

**Line 43: "Conte 2006" should be "Conte et al 2006"**

**We will correct this.**

**Line 96-97: "ITCZ migrates southward during boreal winter" - would be useful to have this marked in Figure 1 too.**

**We will indicate the position of the ITCZ during boreal winter in Figure 1.**

**Line 100: Insert abbreviation (SEC) after South Equatorial Current.**

**We will insert this abbreviation.**

**Line 116: replace "/" with either a space or comma.**

**We will correct this.**

**Line 119: "as result" should be "as a result"**

**We will correct this accordingly.**

**Line 125: "latitudinal transect" is a transect across latitudes. What you have is a "longitudinal transect", i.e. with sites spanning longitudes at a fixed latitude (∼12∘N). C2**

**Thank you for this correction, we will revise this accordingly.**

**Line 183-184: Varved sediments have annual resolution. Since you mentioned "annually to decadally resolved climate records", do you mean "laminated sediments" instead?**

**Yes, this is correct. We will change this accordingly.**

**Line 224: "weight sub-aliquots" is confusing. Suggested rephrasing "sub-aliquots (by weight)".**

**We will rephrase this accordingly.**

**Line 237-238: Confusing sentence. Sounds like you analyzed both ketone and GDGT fractions by both GC and GC/MS - which is likely not the case. Please rephrase.**

**We will rephrase this.**

**Line 285-287: Technically this is a variant of the original BIT index proposed by Hopmans et al 2004. Please rephrase the paragraph to reflect this.**
**Correct. We will emphasize this.**

**Line 296: This is not the first time GC is mentioned in the manuscript. Spell out "gas chromatograph" at the first mention instead of here. Also, there is no need to define the abbreviation at each mention.**

**We will correct this.**

**Line 308-309: Tierney and Tingley (2018) is not the first to notice the warm-end limit of UK'37, i.e. an issue which has been in debate since the 90s. Please include the original references.**

**We will add the original references such as Conte et al (1998), Goñi et al. (2001) and Sicre et al. (2002).**

**Line 313: "gas chromatograph (GC)" see comment to Line 296.**

**We will correct this.**

**Line 314: "mass spectrometer (MS)" see comment to Line 296.**

**We will correct this.**

**Section 2.5 Time-series analysis: Since the result of the time-series analysis is not a main part of the results and discussion, I would suggest to either (A) remove this rather long section or (b) move it to the supplement and add a supplementary figure depicting the result (which is briefly discussed in the text but not shown).**

**We agree, and we will move the methods and results with respect to the time-series analysis to the supplement.**

**Section 3. Results: I had a hard time going through the 4-page long results section. Given the large data set spanning several sites and including several biomarkers and their associated proxies (for which I applaud the authors), this is perhaps inevitable. But I think that it will make the section more accessible for the reader if the authors C3 could reduce the text by 10 to 20%, either by restructuring**

the text, tabulating some of the results and/or limiting the result description to only the main findings that are discussed in the following section.

**We agree that the results section is a bit on the long side , and we will try to shorten it.**

**Line 362: "longitudinal" not "latitudinal".**

**We will correct this accordingly.**

**Line 368-369: Confusing. Rephrase please.**

**We will rephrase this sentence.**

**Line 430-431: "during January and July" - replace with "between January and July". Also, it is not clear at all in Fig 5d that the TEX86H temperatures are lower during these months. Please rephrase.**

**We will rephrase accordingly, and we agree that for M4 this decrease in $TEX_{86}^{H}$ temperatures is not clearly visible and we will remove this statement.**

**Line 444: I'd argue that there's some structural similarity between the Diol Index and chlorophyll-a records.**

**We do not believe this (visual) agreement is strong enough to make a statement about this. Therefore, we would like to refrain from discussing this.**

**Line 482: What are "15 and 18∘ "? Latitude?**

**We will add the latitude.**

 #Line 491-497: I strongly urge the authors to at least show the wavelet analysis in the supplementary info to support their claim. Please also mark the cool water events in Figure 8b to support the claim that ". . .the timing of the observed time periods of enhanced Diol Index variability are similar to those of the cool water events. . ."

**We will show the wavelet analysis in the supplements. However, we cannot mark the cool water events in Fig 8b since we do not know the timings of these events for this specific time interval. We merely wanted to emphasize that Malauene et al. (2014) reported bimonthly frequency and a boreal winter timing for these cold events, which we also observe in our wavelet analysis. We will clarify this. Below are the wavelet results which we will include in the supplements:**

[Figure]

**Fig. S1. a)** The local wavelet power spectrum of the Diol Index in the sediment traps of the Mozambique Channel using the Morlet wavelet, normalized by the standard deviation. On the *x*-axis is time, and the *y*-axis shows the Fourier period in days. The shaded contours are at normalized variances of 0.25, 0.5, 1, 2, and 4. The bold red contour encloses regions of greater than 95% confidence for a red-noise process with a lag-1 coefficient of 0.72. Regions below the dotted red curve are where edge effects become important (Torrence and Compo, 1998). **b)** Global wavelet spectrum of Diol Index – the wavelet spectrum averaged in time over the whole time series. The red dashed line is the 95% confidence level. **c)** Wavelet power averaged over the range of scales from 42 to 90 days. The black line is the time series of the average variance within the 42-90-day range. The red dashed line is the 95% confidence level.

**Line 496-497: I am not following this. Assuming a sampling interval of 21 days - that would give us about 21 data points per year. With so few data points in the time series, it would be impossible to detect 4 cycles in the first half of 2006. Please clarify.**

**With a sampling interval of 21 days, the highest frequency we can detect is half the sampling rate, i.e. 1/42 cycles per day (or 8.7 cycles per year). As we describe on line 492 – 494, and now show in figure S1, the wavelet analysis showed significant variability at about bimonthly frequency (60-day period) during some parts of the time series, most notably the first half of 2006. We rephrase the sentence on line 496 – 497 to: "The strongest variability of the Diol Index at about bimonthly frequencies occurred in the first half of 2006."**

**Line 498-499: It would be helpful to mark the timing of the passage of eddies in Fig 8b.**

**This is a good suggestion; however, it is not completely straightforward to do this in a thorough way. We first need to decide on a definition of a passing eddy – there are several possibilities, for**

**example using the instrumental records of temperature, salinity, or current velocity at the moorings (one useful criterion could be, for example, lateral velocity shear between the eastern and western side), or an independent record such as dynamic height derived from satellite altimetry. Because of this uncertainty we refrain from indicating this.**

**Line 504: "Fig 5c"shows LDI not Diol Index.**

**We will correct this.**

**Line 508: "Fig 5e" shows LDI not Diol Index.**

**We will correct this.**

**Line 522: Change "due its closer vicinity" to "due to its closer vicinity"**

**We will correct this accordingly.**

**Line 523: "NW Africa" This is mentioned only once in the text. Spell out NW. C4**

**We will correct this.**

**Line 556: r (and p) values are more appropriate as a metric to describe the correlation between two variables than r2 (which is used to describe how well the data fit the linear regression model).**

**We will mention the *r* and *p* values here.**

**Line 570-571: Explain briefly why one can expect LCD and levoglucosan to have similar response to degradation, e.g. in terms of their chemical behavior/structure.**

**Both are functionalized polar lipids with alcohol groups and thus are chemically relatively similar. Compared to e.g. fatty acids (carboxyl group) or *n*-alkanes (no functional groups) they are expected to have relatively similar degradation rates.**

**Line 578: "for" or "in" the Atlantic?**

**We will correct this.**

**Line 583-586: Include in the sentence the producers of 1,13- and 1,15-diols.**

**We will correct accordingly.**

**Line 614: Replace "minimal differences" with "minimal variations/variability".**

**We will correct this accordingly.**

**Line 625-627: It is true for LDI and UK'37 that the difference between proxy temperatures and instrumental SST increase during the warmer months, but not for TEX86H. The difference between TEX86H and SST for the cooler months are almost as large as that during the warmer months, and these differences are within the calibration error. Please rephrase the sentence to reflect this.**

**We will rephrase this accordingly.**

**Line 638-640: Taken into account proxy uncertainty, I do not think it is possible to discern if the LDI temperatures are closer to SST or 20m (some temperatures are even higher than SST!), as the isotherms of the upper 30m are so close to each other anyway during the upwelling season. In any case, a habitat depth of the upper 20m is consistent with previous studies as well (as mentioned in line 646 - 649). Please rephrase the sentence.**

**We agree, and we will indeed emphasize that temperature differences are within calibration error and we will rephrase this more nuanced.**

**Line 676-690: This discussion is not supported by the < 2 °C of temperature difference between TEX86H and satellite-SST that is well within the calibration error of TEX86H. In fact, the difference is even smaller than that between the LDI temperature and satellite SST in the North Atlantic (Fig 5), which the authors did not discuss since the differences are mostly within the calibration error. The authors also need to justify why they compared the 0-150m (instead of from the same water depths as the calibration) temperatures with the temperature estimates calculated using the 0-200m calibration. Since the focus of the paper is on LCD proxies, and this subsurface TEX86 finding was not mentioned in the abstract nor the conclusions, I would suggest to remove this paragraph.**

**We agree with the referee that this discussion is outside the scope of this manuscript, and that indeed we are discussing temperature differences which are within calibration error. We therefore will remove this part of the discussion.**

**Line 700-703: See comment on #Line 638-640.**

**We will tone down this statement in terms of the proxy uncertainty.**

**Fig. 2: It took me a while to understand this figure. I think stacked bar chart would make a better option here, so instead of 12 panels with 3 bars each, you'd have 12 stacked bars which give you the same amount of information.**

**We will create a stacked bar chart in the revised version of this manuscript.**

**Line 1184: Change "concentration" to "concentrations".**

**We will correct this accordingly.**

**Line 1185: Change "than" to "then".**

**We will correct this accordingly.**

**Fig. 3: It is impossible to tell which lines/variables correspond to which y-axes without going through the caption. I would suggest to change the color of the right y-axis and its label (Total mass flux) to grey, i.e. the same color as the plot for the variable.**

**We agree, and we will adjust the figures accordingly.**

**Fig. 4: This figure is mentioned for the first time at line 5XX in the section "Discussion" - I suggest to renumber it according to the order of its appearance in the text.**

**We will rearrange the order of figures.**

**Fig. 7: Specify at least in the caption if the annual mean WOA SST is averaged over latitudes or at a fixed latitude. I would also remove the panel on the left and the annual mean T0-150m in panel d if line 676-690 are removed.**

**The annual mean WOA SSTs are specific for the coordinates of the surface sediments; we will emphasize this more.**
**Since we will remove the discussion part on the subsurface TEX$_{86}$, we will remove the left panel (a) and the annual mean T0-T150m and TEX$_{86}$-subsurface temperatures in panel d.**

---

## Author Response (AR2)

**Response to reviewers**

**Submission bg-2019-15 to *Biogeosciences**

**Editor**

Dear Marijke et al.,

I have now received two reports on your contribution. Both find your manuscript suitable for publication in BG following (minor) revisions. Please follow their detailed comments closely in revising your ms.

Sincerely,

Markus

**We thank the editor for the positive assessment of our manuscript. We have revised the manuscript and below we provide point-to-point answers to the comments of the reviewers: when applicable, we indicated where adjustments were made in the text (note: when we refer to line numbers in which we have made adjustments, we refer to the line numbering of the revised manuscript with "track changes"/All Markup). The reviewers' comments are in regular font; our replies are in bold font.**

**Sincerely, also on behalf of all co-authors,**

**Marijke de Bar**

**Response to referee #1**

Review: This manuscript investigates long-chain diols (LCDs) in sediment trap time series from five tropical sites (tropical North Atlantic, Cariaco Basin, Mozambique Channel) to assess seasonal variations in fluxes of LCDs and associated proxies (Long chain Diol Index and Diol Index). These data are compared with other lipid proxies (alkenones and GDGTs) and previous published data (primary production, SST,..). Results show that surface sediment LDI temperatures in the Atlantic and Mozambique Channel compare well with the average LDI-derived temperatures from the overlying sediment traps, as well as with decadal annual mean SST. In the Mozambique Channel and the tropical Atlantic, the LDI temperatures reveal minimal seasonal change although there are clear seasonal SST contrasts, which is likely due to lateral advection of re-suspended sediment. In the Cariaco Basin, a strong seasonality in the LDI is observed, which is linked to the upwelling season and stratification of the water column. In addition, in the Atlantic, the Diol Index reflects a pre-upwelling signal, whereas in the Cariaco Basin, the Diol Index seems to be an indicator of upwelling intensity. This paper is a valuable contribution to the understanding of the seasonal production of LCDs in marine environments and how it is translated in the temperature proxy LDI and the Diol Index (upwelling proxy). A strength of the paper is that the LCD data has been compared with other available data for each site (primary production, SST, alkenones, GDGTs,…), which gives a broader picture and supports the interpretations based on LCDs. The writing style is clear and precise and the interpretations are generally supported by the data. This manuscript is thus suitable for Biogeosciences. However, the current manuscript could be improved before publication. Please find my comments below.

**We thank the referee for the positive assessment and for the comments, which we will discuss below.**

General comments:

Diol index and upwelling: The authors argue that, in the Cariaco Basin, the Diol Index is an excellent indicator of upwelling intensity (Lines 476-480). However, when looking at the 1999-2000 time series, high values of the diol index actually occur when the primary production decreases. What are the R2 values (and p values) that justify "a strong correlation with primary production rates"?

**We agree with the reviewer that for the 1999-2000 time series there is a disagreement during January/February when the diol index increases and primary production rates decrease. We now mention this in the revised version of the manuscript. The 'strong correlation' between the diol index and primary production is based purely on the visual agreement between both time series. We were not able to perform a correlation analysis since the data are differently spaced in time. We have also emphasized this in the revised version manuscript (lines 493-496):**
**"*In the Cariaco Basin, the Diol Index shows a strong correlation (visually as correlation analysis was not possible due to differently spaced data in time) with primary production rates, suggesting that Proboscia productivity was synchronous with total productivity (Fig. 8), although for the 1999-2000 time series there is a disagreement during January/February.*"**

In addition, for the eastern Atlantic (M1 trap), the authors argue that the Diol Index reflects a preupwelling signal, consistent with the current knowledge on Proboscia ecology (Lines 509-526). I would like to see more discussion that explains why at one location the Diol index indicates preupwelling conditions, whereas it seems to be an indicator of upwelling intensity at another location.

**We agree that this seems contradictory and requires more discussion, which we have implemented in the revised version of the manuscript. The Diol Index is an upwelling indicator based on the assumption that *Proboscia* diatoms generally thrive in upwelling regions. However, the index is in fact an indicator for *Proboscia* productivity, and whether it reflects upwelling/pre-upwelling/stratification/etc. conditions will depend on the region and the local ecological dynamics determining the role of *Proboscia* diatoms (e.g., Rampen et al., 2014; de Bar et al., 2018). Studies have shown that *Proboscia* diatoms are often more dominant during early/pre-upwelling because they need relatively little silica and they are able to migrate to deeper waters to obtain nutrients (Koning et al., 2001) and sediment trap data from Wakeham et al. (2002), Prahl et al. (2000), Sinninghe Damsté et al. (2003) and Rampen et al. (2007) show that *Proboscia* lipids (diols and/or hydroxyl methyl alkanoates) are highest during early upwelling. Therefore, we hypothesize that this Diol Index maximum during spring which we observe for station M1 in the Atlantic might be a pre/early-upwelling signal since the upwelling in the Guinea Dome often occurs between July and October (Siedler et al., 1992). Indeed, *Proboscia* diatoms do not reflect early-upwelling in every region. Reports of *Proboscia* spp. blooms vary from stratification to early-upwelling to postbloom, and from high nutrients to low nutrients (see Rampen et al., 2014; references in Table 1). Apparently, in the Cariaco Basin, *Proboscia* diatoms bloom relatively synchronous with general productivity, as evidenced from the agreement between the Diol Index and primary production time series, emphasizing the value of sediment trap studies like ours in revealing regional differences in proxy signals. We have added the following lines (546-549):**

**"*Our results clearly show that the Diol Index reflects different things in different regions. This is due to the ecology of Proboscia spp. where blooms occur during stratification to early upwelling to postbloom, and from high nutrients to low nutrients (see Rampen et al., 2014; references in Table 1). Therefore, the type of conditions reflected by the Diol Index is specific for every region.*"**

Keto-ols as oxidation products (Lines 578-586): An alternative explanation for the non-detection of 1,14-keto-ols would be that keto-ols are not oxidation products of LCDs, but rather produced by unknown organism(s) (Versteegh et al., 1997). Previous studies have indeed shown the absence of evidence of conversion of diols into their corresponding oxidized keto-ols (Jiang et al., 1994; Méjanelle al 2003; Shimokawara et al., 2010). I think the authors should acknowledge this.

**We agree, and we have mentioned this hypothesis as well (lines 611-615):**
**"*Alternatively, the keto-ols are not oxidation products but are produced by unknown organisms in the water column. In fact, Méjanelle et al. (2003) observed trace amounts of $C_{30}$ 1,13- and $C_{32}$ 1,15-keto-ols in cultures of the marine eustigmatophyte Nannochloropsis gaditana. Thus, an alternative explanation for the non-detection of 1,14-keto-ols is that in contrast to the 1,15-keto-ols, they were not produced in the water column.*"**

Figures: I think the current order of the figures does not necessarily follow the logic of the results/discussion. For more clarity, I would suggest modifying the order as follows: Fig. 2 should be Fig. 8; Fig. 3 should be Fig. 2; Fig. 8 should be Fig. 8; Fig. 4 should be Fig. 3; Fig. 5 should be Fig. 8; Fig. 6 should be Fig. 4; Fig. 8 should be Fig. 5; Fig. 8 should be Fig. 6.

**We have re-ordered as follows:**
**Fig. 2 → Fig. 2**
**Fig. 3 → Fig. 3**
**Fig. 4 → Fig. 9**
**Fig. 5 → Fig. 4**
**Fig. 6 → Fig. 5**
**Fig. 7 → Fig. 6**
**Fig. 8 → Fig. 7**
**Fig. 9 → Fig. 8**

Specific comments:

Line 25: specify "with emphasis on the temperature proxy Long Chain Diol Index".

**We have corrected this accordingly.**

Line 27: specify "similar to the two other lipid-based temperature proxies TEX86 and UK'37".

**We have corrected this accordingly.**

Line 27: "In addition" instead of "However".

**We have corrected this accordingly.**

Line 29: Could be rephrased as: "In contrast, the LDI in the Cariaco Basin shows larger seasonal variation".

**We have corrected this accordingly.**

Line 48: Need references.

**We have added the review of Tierney (2014) as reference.**

Lines 48-50: Could be rephrased as: "However, research showed that despite their highest abundance being recorded in the upper 100 m of the water column, Thaumarchaeota can be present down to 5000 m depth (Karner et al., 2001; Herndl et al., 2005)".

**We have corrected this accordingly.**

Line 69: "for autumn to summer" should be "for autumn and summer".

**We have corrected this accordingly.**

Figure 1: indicate in the caption what NEC, NECC, SEC, MC, GD, NBC and GC stand for. Is it possible to add the position of the ITCZ during the boreal winter?

**We have clarified the abbreviations in the figure caption and indicated the position of the ITCZ during boreal winter.**

Line 200: What are CTD measurements?

**We refer here to temperature measurements of seawater at 1m water depth sampled by CTD. We have clarified this.**

Line 256-258: Could diols be found in the DCM:MeOH (1:1; v/v) fraction? Have you checked?

**We thank the reviewer for noticing this, since the sentence is incorrect: not the MeOH fractions were analyzed for diols, but the DCM:MeOH (1:1, v/v) fractions. We have corrected this.**

Line 369: Should be as: "C28 and C30 1,13- (0–3 %), the C30 1,15- (44–99 %), and the C32 1,15-diols (0–7%)".

**We have corrected this accordingly.**

Lines 367-376: I think a table showing the presence/absence for each diols (and the % of total LCDs) at the different traps (M1, M2,...) and different sites (Atlantic, Mozambique Channel, Cariaco Basin) would be useful to clearly see which diols are detected for each location. The Figure 2 is used to discuss the preservation between traps and sediments rather than showing the diols detected.

**We do not fully agree, since the number of figures is already extensive, as is the result section, and we consider this relatively detailed.**

Line 392: Fig. 4 is cited before Fig. 8. I think the order of the figures should be changed (see previous comment).

**We have changed the order of figures, see comment above.**

Line 397: cite Figure 7.

**We have corrected this.**

**Response to referee #2**

General comments

In this study, de Bar et al. presented long-chain diol (LCD) data from five sites; three along a longitudinal transect in the tropical Atlantic, the Cariaco Basin and the Mozambique Channel. LCD derived indices, i.e. Long-chain Diol Index (LDI) and Diol Index, are used to reconstruct past SST and upwelling, respectively. These proxies are relatively new compared to those based on alkenones and GDGTs, thus have not been as well studied. This is where this study comes in. de Bar et al analyzed LCDs from sediment traps and underlying sediments. For the sites where alkenones and GDGT data do not yet exist, the authors also analyzed these biomarkers in addition to LCDs C1 to allow multi-proxy comparison for all the sites. The well-designed experiment thus allows the authors to investigate various aspects of the LCDs and their associated proxies, including the temporal evolution (seasons to years), settling processes, as well as comparison with other commonly applied biomarker proxies. The data presented by de Bar et al. generally show that LDI-derived temperatures agree within error with instrumental data in the Atlantic, albeit with different amplitude of change. At upwelling sites, the Diol Index seems to either record a pre-upwelling signal or show the same trend as in primary productivity.

The study fits the scope of Biogeosciences, and will also be of interest to readers from other community such as paleoclimate. The manuscript is generally well-written and accessible. I do, however, feel that some figures could be further improved for clarity. I find the "Results" section too long and some discussion unclear or not fully supported by the data, especially in section 4.3. Below are suggestions and comments that I hope will help the authors in further improving the manuscript. Once the concerns are addressed, I strongly recommend the publication of this manuscript.

**We thank the referee for the positive assessment and for the comments, which we will discuss below.**

Specific comments

**Line 34-36: Clunky sentence. Please rephrase.**

**We have rephrased as follows (lines 35-39):**
**"*Lastly, we observed large seasonal variations in the Diol Index, as indicator of upwelling conditions, at three sites: in the Eastern Atlantic potentially linked to Guinea Dome upwelling, in the Cariaco Basin likely caused by seasonal upwelling, and in the Mozambique Channel where underlying mechanisms are indefinable but where Diol Index variations may be driven by upwelling from favorable winds and/or eddy migration.*"**

**Line 43: "Conte 2006" should be "Conte et al 2006"**

**We have corrected this.**

**Line 96-97: "ITCZ migrates southward during boreal winter" - would be useful to have this marked in Figure 1 too.**

**We have indicated the approximate position of the ITCZ during boreal winter in Figure 1.**

**Line 100: Insert abbreviation (SEC) after South Equatorial Current.**

**We have inserted this abbreviation.**

**Line 116: replace "/" with either a space or comma.**

**We have replaced it with a comma.**

**Line 119: "as result" should be "as a result"**

**We have corrected this accordingly.**

**Line 125: "latitudinal transect" is a transect across latitudes. What you have is a "longitudinal transect", i.e. with sites spanning longitudes at a fixed latitude (∼12◦N). C2**

**Thank you for this correction, we have corrected this throughout the manuscript.**

**Line 183-184: Varved sediments have annual resolution. Since you mentioned "annually to decadally resolved climate records", do you mean "laminated sediments" instead?**

**Yes, we have corrected this accordingly.**

**Line 224: "weight sub-aliquots" is confusing. Suggested rephrasing "sub-aliquots (by weight)".**

**We have corrected this accordingly.**

**Line 237-238: Confusing sentence. Sounds like you analyzed both ketone and GDGT fractions by both GC and GC/MS - which is likely not the case. Please rephrase.**

**We have rephrased as follows (lines 244-247):**
**"*The ketone fraction was also dissolved in ethyl acetate, and analyzed by GC and GC/MS. The GDGT fraction was dissolved in hexane:isopropanol (99:1, v/v), filtered through a 0.45 µm polytetrafluoroethylene (PTFE) filter and analyzed by HPLC-MS.*"**

**Line 285-287: Technically this is a variant of the original BIT index proposed by Hopmans et al 2004. Please rephrase the paragraph to reflect this.**
**We have rephrased as follows (lines 293-296):**
**"*The Branched Isoprenoid Tetraether (BIT) index is a proxy for the relative contribution of terrestrial derived organic carbon (Hopmans et al., 2004). We have calculated the modified version as reported by de Jonge et al. (2014; 2015) which is based on the original index as proposed by Hopmans et al. (2004), but includes the 6-methyl brGDGTs.*"**

**Line 296: This is not the first time GC is mentioned in the manuscript. Spell out "gas chromatograph" at the first mention instead of here. Also, there is no need to define the abbreviation at each mention.**

**We have corrected this accordingly.**

**Line 308-309: Tierney and Tingley (2018) is not the first to notice the warm-end limit of UK'37, i.e. an issue which has been in debate since the 90s. Please include the original references.**

**We have rephrased as follows (lines 316-321):**
**"*We have also applied the recently proposed BAYSPLINE Bayesian calibration of Tierney and Tingley (2018). They and others have shown that the $U^{K'}_{37}$ estimates substantially attenuate above temperatures of 24 °C (e.g., Conte et al., 1998; Goñi et al., 2001; Sicre et al., 2002). The Bayesian calibration moves the upper limit of the $U^{K}_{37}$ calibration from approximately 28 to 29.6 °C at unity. Since our traps are located in tropical regions with SSTs > 24 °C, we have applied this calibration as well.*"**

**Line 313: "gas chromatograph (GC)" see comment to Line 296.**

**We have corrected this accordingly.**

**Line 314: "mass spectrometer (MS)" see comment to Line 296.**

**We have corrected this accordingly.**

**Section 2.5 Time-series analysis: Since the result of the time-series analysis is not a main part of the results and discussion, I would suggest to either (A) remove this rather long section or (b) move it to the supplement and add a supplementary figure depicting the result (which is briefly discussed in the text but not shown).**

**We agree, and we have moved these methods to the supplements.**

**Section 3. Results: I had a hard time going through the 4-page long results section. Given the large data set spanning several sites and including several biomarkers and their associated proxies (for which I applaud the authors), this is perhaps inevitable. But I think that it will make the section more accessible for the reader if the authors could reduce the text by 10 to 20%, either by restructuring the text, tabulating some of the results and/or limiting the result description to only the main findings that are discussed in the following section.**

**We agree that the results section is a bit on the long side, and we have removed a few sentences.**

**Line 362: "longitudinal" not "latitudinal".**

**We have corrected this.**

**Line 368-369: Confusing. Rephrase please.**

**We have rephrased as follows (lines 378-380):**
**"*The LCDs detected in the sediment trap samples and surface sediments from the tropical North Atlantic (Fig. 2) are the $C_{28}$, $C_{30}$ and $C_{30:1}$ 1,14-, $C_{28}$ and $C_{30}$ 1,13-, the $C_{30}$ 1,15-, and $C_{32}$ 1,15- diols.*"**

**Line 430-431: "during January and July" - replace with "between January and July". Also, it is not clear at all in Fig 5d that the TEX86H temperatures are lower during these months. Please rephrase.**

**We have rephrased accordingly, and we agree that for M4 this decrease in $TEX_{86}^{H}$ temperatures is not clearly visible and we have removed this statement.**

**Line 444: I'd argue that there's some structural similarity between the Diol Index and chlorophyll-a records.**

**We do not believe this (visual) agreement is strong enough to make a statement about this. Therefore, we would like to refrain from discussing this.**

**Line 482: What are "15 and 18∘ "? Latitude?**

**We have added the latitude.**

**Line 491-497: I strongly urge the authors to at least show the wavelet analysis in the supplementary info to support their claim. Please also mark the cool water events in Figure 8b to support the claim that ". . .the timing of the observed time periods of enhanced Diol Index variability are similar to those of the cool water events. . ."**

**We now show the wavelet analysis in the supplements. However, we cannot mark the cool water events in Fig 8b since we do not know the timings of these events for this specific time interval. We merely wanted to emphasize that Malauene et al. (2014) reported bimonthly frequency and a boreal winter timing for these cold events, which we also observe in our wavelet analysis. Below are the wavelet results which we have included in the supplements:**

[Figure]

**Fig. S1. a) The local wavelet power spectrum of the Diol Index in the sediment traps of the Mozambique Channel using the Morlet wavelet, normalized by the standard deviation. On the *x*-axis is time, and the *y*-axis shows the Fourier period in days. The shaded contours are at normalized variances of 0.25, 0.5, 1, 2, and 4. The bold red contour encloses regions of greater than 95% confidence for a red-noise process with a lag-1 coefficient of 0.72. Regions below the dotted red curve are where edge effects become important (Torrence and Compo, 1998). b) Global wavelet spectrum of Diol Index – the wavelet spectrum averaged in time over the whole time series. The red dashed line is the 95% confidence level. c) Wavelet power averaged over the range of scales from 42 to 90 days. The black line is the time series of the average variance within the 42-90-day range. The red dashed line is the 95% confidence level.**

**Line 496-497: I am not following this. Assuming a sampling interval of 21 days - that would give us about 21 data points per year. With so few data points in the time series, it would be impossible to detect 4 cycles in the first half of 2006. Please clarify.**

**With a sampling interval of 21 days, the highest frequency we can detect is half the sampling rate, i.e. 1/42 cycles per day (or 8.7 cycles per year). As we describe on line 508-511, and now show in figure S1, the wavelet analysis showed significant variability at about bimonthly frequency (60-day period) during some parts of the time series, most notably the first half of 2006. We have rephrased the sentence on line 516-517 to: "*The strongest variability of the Diol Index at about bimonthly frequencies occurred in the first half of 2006.*"**

**Line 498-499: It would be helpful to mark the timing of the passage of eddies in Fig 8b.**

**This is a good suggestion; however, it is not completely straightforward to do this in a thorough way. We first need to decide on a definition of a passing eddy – there are several possibilities, for example using the instrumental records of temperature, salinity, or current velocity at the**

**moorings (one useful criterion could be, for example, lateral velocity shear between the eastern and western side), or an independent record such as dynamic height derived from satellite altimetry. Because of this uncertainty we refrain from indicating this.**

**Line 504: "Fig 5c"shows LDI not Diol Index.**

**We have corrected this.**

**Line 508: "Fig 5e" shows LDI not Diol Index.**

**We have corrected this.**

**Line 522: Change "due its closer vicinity" to "due to its closer vicinity"**

**We have corrected this accordingly.**

**Line 523: "NW Africa" This is mentioned only once in the text. Spell out NW.**

**We have corrected this accordingly.**

**Line 556: r (and p) values are more appropriate as a metric to describe the correlation between two variables than r2 (which is used to describe how well the data fit the linear regression model).**

**We now mention the *r* and *p* values here.**

**Line 570-571: Explain briefly why one can expect LCD and levoglucosan to have similar response to degradation, e.g. in terms of their chemical behavior/structure.**

**We have included the following (lines 594-596):**
**"*Both are functionalized polar lipids with alcohol groups and thus are chemically relatively similar when compared to e.g. fatty acids (carboxyl group) or n-alkanes (no functional groups).*"**

**Line 578: "for" or "in" the Atlantic?**

**We have corrected this sentence.**

**Line 583-586: Include in the sentence the producers of 1,13- and 1,15-diols.**

**We have corrected this accordingly.**

**Line 614: Replace "minimal differences" with "minimal variations/variability".**

**We have corrected this accordingly.**

**Line 625-627: It is true for LDI and UK'37 that the difference between proxy temperatures and instrumental SST increase during the warmer months, but not for TEX86H. The difference between TEX86H and SST for the cooler months are almost as large as that during the warmer months, and these differences are within the calibration error. Please rephrase the sentence to reflect this.**

**We have added the following (lines 655-657):**
**"*Interestingly, the $U^{K'}_{37}$- and $TEX^{H}_{86}$-derived temperature trends show the same phenomenon (Turich et al., 2013; Fig. 8), where the proxy temperatures are cooler than the measured temperatures during the warmer months. However, in contrast to the $U^{K'}_{37}$ and LDI, the $TEX^{H}_{86}$ also overestimates SST overestimation during the cold months.*"**

**Line 638-640: Taken into account proxy uncertainty, I do not think it is possible to discern if the LDI temperatures are closer to SST or 20m (some temperatures are even higher than SST!), as the isotherms of the upper 30m are so close to each other anyway during the upwelling season. In any case, a habitat depth of the upper 20m is consistent with previous studies as well (as mentioned in line 646 - 649). Please rephrase the sentence.**

**We agree, and we have now emphasized that the temperature differences are within calibration error (lines 669-673):**
**"*During upwelling, LDI-temperatures agree better with SST, implying that the habitat of the LCD producers potentially was closer to the surface, coincident with the shoaling of the nutricline and thermocline (Fig. 10). However, these absolute differences in LDI-temperatures are generally within the calibration error (2 °C), and these seasonal variations in LDI-temperatures should thus be interpreted with caution.*"**

**Line 676-690: This discussion is not supported by the $< 2\ ^0$C of temperature difference between TEX86H and satellite-SST that is well within the calibration error of TEX86H. In fact, the difference is even smaller than that between the LDI temperature and satellite SST in the North Atlantic (Fig 5), which the authors did not discuss since the differences are mostly within the calibration error. The authors also need to justify why they compared the 0-150m (instead of from the same water depths as the calibration) temperatures with the temperature estimates calculated using the 0-200m calibration. Since the focus of the paper is on LCD proxies, and this subsurface TEX86 finding was not mentioned in the abstract nor the conclusions, I would suggest to remove this paragraph.**

**We agree with the referee that this discussion is outside the scope of this manuscript, and that indeed we are discussing temperature differences which are within calibration error. We therefore have removed this part of the discussion.**

**Line 700-703: See comment on #Line 638-640.**

**We have rephrased as follows (lines 732-736):**
**"*In the Cariaco Basin we observe a seasonal signal in the LDI linked to the upwelling season reflecting temperatures of the upper ca. 30 m of the water column.*"**

**Fig. 2: It took me a while to understand this figure. I think stacked bar chart would make a better option here, so instead of 12 panels with 3 bars each, you'd have 12 stacked bars which give you the same amount of information.**

**We have tried this option, but to our opinion this did not improve clarity as it visually suggests that the preservation percentages are summed. We therefore chose to use our original figure.**

**Line 1184: Change "concentration" to "concentrations".**

**We have corrected this accordingly.**

**Line 1185: Change "than" to "then".**

**We have corrected this accordingly.**

**Fig. 3: It is impossible to tell which lines/variables correspond to which y-axes without going through the caption. I would suggest to change the color of the right y-axis and its label (Total mass flux) to grey, i.e. the same color as the plot for the variable.**

**We agree, and we have adjusted the figure accordingly.**

**Fig. 8: This figure is mentioned for the first time at line 5XX in the section "Discussion" - I suggest to renumber it according to the order of its appearance in the text.**

**We have re-ordered the figures, also on suggestion of referee#1.**

**Fig. 6: Specify at least in the caption if the annual mean WOA SST is averaged over latitudes or at a fixed latitude. I would also remove the panel on the left and the annual mean T0-150m in panel d if line 676-690 are removed.**

The annual mean WOA SSTs are specific for the coordinates of the surface sediments; we have now emphasized this in the caption.
Since we have removed the discussion part on the subsurface $TEX_{86}$, we have also removed the left panel (a) and the annual mean $T_0$-$T_{150}$m and $TEX_{86}$-subsurface temperatures in panel d.

**Additional comment:**

We have replaced Fig. 9 since we by accident previously plotted the summed 1,13-/1,15-diol concentrations instead of the summed flux-weighted 1,13-/1,15-diol concentrations.

[revised manuscript text omitted]